# A Bibliometric Analysis and Systematic Review on E-Marketplaces, Open Innovation, and Sustainability

Jose Alejandro Cano [1,*], Abraham Londoño-Pineda [1], Maria Fanny Castro [2], Hugo Bécquer Paz [2], Carolina Rodas [1] and Tatiana Arias [1]

[1] Faculty of Economic and Administrative Sciences, Universidad de Medellin, Medellin 050026, Colombia; alondono@udemedellin.edu.co (A.L.-P.); crodas103@soyudemedellin.edu.co (C.R.); tarias249@soyudemedellin.edu.co (T.A.)

[2] Faculty of Economics, Administrative and Accounting Sciences, Universidad Libre de Colombia, Cali 760031, Colombia; mariaf.castroa@unilibre.edu.co (M.F.C.); hugob-pazq@unilibre.edu.co (H.B.P.)

* Correspondence: jacano@udemedellin.edu.co

**Abstract:** In recent years, the rise of e-commerce has prompted the emergence of electronic marketplaces, or e-marketplaces, which act as intermediaries in the buying and selling process, bringing together several vendors to offer a wide range of products and services to customers, generating modalities such as business-to-business (B2B), business-to-consumer (B2C) or consumer-to-consumer (C2C) e-marketplaces. E-marketplaces offer advantages such as access to potential buyers, business and product visibility, the reduction of transaction costs, the comparison of offers and prices among competitors, and the ease of business internationalization. However, the success of e-marketplace business models depends on the sustainability of these platforms, which must involve different stakeholders in order to meet economic, environmental, and social objectives. Therefore, this study presents a bibliometric analysis and systematic review of e-marketplaces, open innovation, and sustainability for the last ten, five, and two years. The analysis includes the number, types, and subject areas of documents published each year, as well as considerations such as the most-cited publications and the leading authors, journals, countries, and institutional affiliations. The analysis also includes a study of the relevant concepts in the publications and their relationships, identifying the predominant topics related to e-marketplaces, open innovation, and sustainability. The results indicate a focus on subject areas such as social sciences, environmental sciences, energy, business, management, and accounting, which is consistent with the economic, environmental, and social dimensions of sustainability. The findings show that e-marketplaces, open innovation, and sustainability are closely related to concepts such as sustainable development, e-commerce, digital marketing, China (the leading country in terms of publications in all periods), logistics, supply chain management, big data, planning, and decision making. Future works should address traffic congestion and environmental impact, new delivery practices in last-mile logistics, and the motives for users' engagement in e-marketplaces. Likewise, future research can be oriented toward sustainability dimensions and stakeholders' integration through open innovation and toward the limitations of SMEs in order to access and benefit from digital platforms.

**Keywords:** sustainability; open innovation; e-marketplaces; bibliometric analysis; Scopus; systematic review

## 1. Introduction

The current economy presents worrying scenarios for multiple sectors due to the recession generated by the COVID-19 pandemic, which caused social confinements worldwide [1,2]. As a result of COVID-19's impact, consumers are increasingly turning to online purchases, and companies, especially Small and Medium Enterprises (SMEs), need to take advantage of the opportunities generated by this crisis through online migration, the

acceptance of electronic commerce, and online payments in order to overcome supply and demand inefficiencies [3]. This situation has prompted information and communication technologies (ICTs) enabling business models based on e-marketplaces for SMEs [4], contributing to the revitalization, dissemination and sustainability of industries [5], facilitating a sustainable mode of urbanization, and enhancing delivery services to urban stakeholders [6].

Likewise, the need has arisen for all industries to implement innovative and sustainable processes that are environmentally friendly [7] and promote sustainable consumption through e-commerce platforms in order to provide intelligent information and satisfy customers and providers [8], and to increase health and safety for the community [9]. In this sense, e-marketplaces offer opportunities to markets by reducing marketing expenses [10], increasing user traffic to perform transactions, affecting consumer purchasing decisions [11], and increasing sales [12]. Moreover, e-marketplaces promote sustainable consumption habits by mitigating travel costs and the carbon footprint of commerce [13]. E-marketplaces represent digital infrastructures that offer products online, allowing buyers and sellers to quickly connect in order to coordinate and satisfy their demands [3,14], and allowing consumers to shop online easily, at any hour of the day, using secure payment systems [15]. In this way, sellers and owners of digital platforms provide the products, consumers buy the products through the platform's website, and logistics companies deliver the product to the consumer [16].

Therefore, e-marketplaces allow commercial transactions through the exchange of commercial information, the maintenance of commercial relations, the achievement of commercial negotiations, and the settlement and the execution of agreements through electronic means using the Internet [17–19]. However, SMEs do not usually participate in electronic marketplaces because they do not perceive the benefits of participating in e-marketplaces [20], considering that trust and information quality represent the determining factors of purchase intention in an e-marketplace [21]. Similarly, consumers and retailers face several sustainable online consumption barriers, such as backward sustainable production technology, similar types of products in offline stores, a lack of information about a product when shopping online, a lack of policy support, a lack of government regulations, a lack of awareness of sustainable consumption, and a reduced level of costumers' consumption [22]. Consequently, e-marketplaces must implement open innovation processes that allow knowledge of the expectations, perceptions, and objectives of stakeholders (suppliers, carriers, logistics providers, users, and the government) [23], either in B2B business models [24], B2C business models [25], or C2C business models [26]. This leads to the design of sustainable business models that benefit e-marketplaces, society, and the environment in a profitable way [27]. Therefore, open innovation would impact the design of business models by inspiring stakeholders to participate in electronic marketplaces [28].

The traditional approach of innovation depends on ideas generated and developed within R&D departments, or on the hiring of experts for business units requiring innovation activities [29]. Contrary to this approach, the concept of open innovation implies the dynamic interaction of companies with a wide variety of external stakeholders such as customers, users, suppliers, universities, research and development centers, competitors, government agencies, neighboring communities, and society in general [30]. Therefore, open innovation results from the integration of knowledge, information, research, development, and innovation provided by stakeholders to satisfy the needs and desires of the market with the appropriate technology and resources, assuming that knowledge that promotes innovations can be found anywhere in the value chain [31,32]. Hence, following a collaborative or cooperative approach, people, organizations, and communities represent active participants in the co-production of goods and services [33].

The open innovation approach stresses the importance of the external knowledge activity of a company during the innovation process [34], and it allows the creation of value and the advancement of technologies through the inflows and outflows of knowledge of different companies through new combined business models [35,36]. As such, many companies

and suppliers (stakeholders) join digital platforms in order to cooperate and compensate for the lack of innovation resources within a particular company, thus improving the level of innovation, the existing capacities, and the speed of market development. [30,37,38]. Therefore, open innovation promotes collaborative design processes based on a network system between companies in order to foster user innovation and increase the ability to respond quickly to changing markets [35,39].

Open innovation in e-marketplace platforms can be achieved through collaboration between the platform owner and physical intermediary firms, in which the platform interacts with logistics companies to make joint use of information in order to improve delivery efficiency [16]. Likewise, open innovation processes in e-marketplaces must include their main stakeholders—such as supplier companies and users—in order to design inclusive business models that contribute to the e-marketplace value and the achievement of economic, social, and environmental goals [40,41]. Likewise, the consideration of the different stakeholders in an e-market business model is vital because its viability depends on its ability to attract a sufficient number of participants; then, as more individual firms in an industry join a marketplace, its currency or worth within the industry will rise, encouraging more firms to participate [42,43]. Therefore, it is necessary to highlight the potential benefits of e-marketplaces which can attract numerous small- and medium-sized enterprises (SMEs) for the development of sustainable electronic businesses [43,44], and to understand that open innovation will lead to a new dynamic economy and sustainable development [34].

Digital media save time and money in the information exchange with stakeholders, accelerate processes by generating positive experiences for customers, and provide lasting relationships with customers [45]. Likewise, the involvement of stakeholders could help companies overcome market failures and provide specific information and knowledge to innovate in processes, products, and services, thus achieving the economic and sustainability innovation goals simultaneously [32,46]. Consequently, e-marketplaces can be a solution to break the long distribution chain for products [47], and to perform effective processes that optimize times and increase service quality, generating benefits for all stakeholders, involving social goals, customer satisfaction, the reduction of $CO_2$ emissions, and the care of the environment [16,17].

Sustainable urban systems can result from the interaction between stakeholders in the design and implementation of urban consolidation centers, alternate delivery locations (lockers/stores), and zero-tailpipe-emission vehicles. They also allow the implementation of operations and practices that optimize rush deliveries, customer basket size, vendor consolidation levels, trip length and trip frequency, the routing of van deliveries, and the energy efficiency of shop and e-fulfillment center operations [48,49]. Therefore, collaboration is required, and all stakeholders in the virtual market must take appropriate responsibility to protect the environment, preserve natural resources, and maintain and sustain the economy [15].

Moreover, the consistent growth of B2C e-commerce transactions brings with it the negative externalities of increased congestion and pollution due to the increase in trucks entering cities [48], for which different delivery concepts such as bike deliveries or delivery points can benefit either the companies or the quality of life in the city. In this way, operational costs can be reduced by stimulating customer self-pickup, while externalities decrease with the cargo bike distribution system [50]. Environmental sustainability, from a logistics perspective, is related to transportation planning and management, warehousing, packaging, and distribution network design, which can be measured through indicators of energy use, gas emissions, waste generated, and traffic mileage, and involve the implementation of green initiatives such as the use of alternative vehicles, as well as the use of more recent and less polluting vehicles [25].

For B2B e-commerce, a blockchain solution can provide all of the participants in the sustainable B2B buying process with the same data about the trade. It creates a decentralized and secure database that increases the payment speed and the reliability and transparency

of the data transfer [51]. Likewise, the perceived value in B2B e-marketplaces mediates trust in the commodity information and online purchase intention of the procurement personnel [52]. On the other hand, the ownership structure of the e-market (neutral or biased), the type of market competition faced by both the participating firms and the market operator (an oligopoly market or an oligopsony market), and low e-market connection costs to attract firms' participation represent key factors leading to a sustainable e-marketplace [53]. Additionally, multiple factors influence the sustainability of e-marketplaces at the macroeconomic, regulatory, industrial, and individual firm levels. These factors are related to regulation, the position within the economic cycle, the power of buyers and suppliers, the characteristics of the product, industry information technology readiness, strategic intent, and culture [42]. On the other hand, the central relationships between electronic markets and sustainability can be grouped into five clusters: the economic dimension (the viability of electronic markets, and a compelling value proposition to the stakeholders), the environmental dimension (the scarcity of natural resources), the social dimension (focusing on the human side with social capital and social equity), the technological dimension (green information technologies and the reliable and fault-tolerant operation of information systems), and the systemic dimension (the interaction of many actors reflecting social patterns) [14].

Consequently, sustainability can be achieved through the joint effort of resource and knowledge sharing, aiming for a long-term impact on the economy, the environment, and society [54], confirming that open innovation is essential in order to increase a competitive advantage and improve the response capacity of the organization in order to satisfy the demand for market orientation [55]. Likewise, open innovation facilitates effective strategic sustainable management by promoting sustainable innovations for organizational sustainability [36]. Through open innovation, companies can leverage knowledge management to promote sustainable innovations that influence organizational sustainability [31], and to make the best practices available to everyone in order to optimize the use of resources, respect the environment, and include community values [36].

Based on the abovementioned factors, it is necessary to analyze the role of open innovation in e-marketplaces in order to guarantee the sustainability of business models, providing a balance of supply and demand for natural resources, and prioritizing care for the environment [7]. This involves the consideration of economic, social, and cultural contexts in order to create, capture, and deliver value [27]; these sustainable business models can provide marketing advantages for sustainable products even if they are related to higher prices [56]. On the other hand, opportunities exist to conduct a more detailed investigation of open innovation practices and their suitability for sustainability innovation [46], to identify the ways in which organizations are positioning themselves to develop sustainability with the support of open innovation [31], and to analyze the joint study of sustainability and open innovation supporting the achievement of industry objectives and sustainability goals [57].

Longitudinal studies are relevant for the tracking of changes over time in a specific field, the assessment of phenomena occurring over a long period, and the description of the ways in which perspectives change over time [58,59]. Longitudinal studies applied to scientific literature analysis may require the review of the changes presented in a research field in the last ten (long term), five (medium term), and two (short term) years. The periods may not be consistent when performing longitudinal studies considering the latest ten, five, and two years, as a notable growth of publications in the literature is usually generated in the recent years (i.e., the last two years). Likewise, an analysis of the previous two years allows the identification of the latest trends, opportunities, and research gaps [57].

However, to the best of our knowledge, there are no studies that analyze the progress of research on open innovation and the sustainability of e-marketplaces, establishing elements such as the growth of publications, study areas, leading journals, leading authors, documents by country/territory, leading institutions, the most cited documents, and leading research topics around this subject. Therefore, this article presents a bibliometric

analysis and systematic review on sustainability, open innovation, and e-marketplaces in order to identify research trends on these topics for the long, medium, and short terms. The remainder of this paper is organized as follows. In the next section, we introduce the methodology used to search, collect, and analyze bibliometric information. Section 3 reports the main findings for the long, medium, and short terms. Subsequently, Section 4 discusses the results of the research. The final section presents the conclusions and the research limitations identified.

## 2. Methodology

This article is based on bibliometric analysis and a systematic review of the literature in order to identify, evaluate, and synthesize all of the relevant studies from the existing literature related to e-marketplaces, open innovation, and sustainability. The source of information for this study was the Scopus database, to which the search equation TITLE-ABS-KEY (("Open innovation" AND Sustainab*) OR (Sustainab* AND (e-marketplace* OR e-commerce OR "digital market*")) OR ("Open innovation" AND (e-marketplace* OR e-commerce OR "digital market*")) OR ("Open innovation" AND Sustainab* AND (e-marketplace* or e-commerce or "digital market*"))) was applied, ensuring that the most relevant documents that address the issues of e-marketplaces, open innovation, and sustainability are covered simultaneously or in pairs of concepts. The documents obtained with the search equation were filtered into three periods: 2012–2021 (long term), 2017–2021 (medium term), and 2020–2021 (short term).

Therefore, this document studies, in the literature, the evolution of concepts in the last ten, five and two years, identifying changes and trends in research around the chosen topic. For each period, this study analyzes the growth in publications, main subject areas, leading journals, leading authors, documents by country/territory, institutional affiliation, most-cited documents, main research topics addressed, and concept co-occurrence. For the growth in publications, trend graphs are used for the number of documents in the literature, establishing whether the topic addressed is relevant to the scientific community in recent years. The study of the subject areas establishes the focus received by publications on e-marketplaces, open innovation, and sustainability, and allows us to understand how this topic is addressed in the literature.

The analysis also focuses on the journals and authors with the highest number of published documents, pointing out the recommended sources for publishing and the authors who contribute the most in the subject of study. The study of documents by country/territory and institutional affiliation determines the geographical locations and the institutions with the highest interest in the subject of study. The analysis of the most-cited documents highlights the specific research topics around sustainability, open innovation, and e-marketplaces. On the other hand, the most-used keywords in the collected documents allow the identification of the main concepts associated with the study topic, and the co-occurrence of these concepts is established using the VOSviewer software. This software groups the concepts into clusters, and the number of clusters depends on the number and type of concepts in each period. Each cluster has a central node gathering around many concepts, and they appear in large spheres in the co-occurrence graph.

## 3. Results

### 3.1. Long-Term Bibliometric Analysis: 2012–2021

According to the Methodology section, the evolution of Scopus publications on e-marketplaces, open innovation, and sustainability in the last ten years shows exponential growth, generating a total of 1524 documents. According to Figure 1, the highest growth rates in publications compared to the previous year were 48.8%, 48.6%, and 46.4% for 2016, 2019, and 2021, respectively.

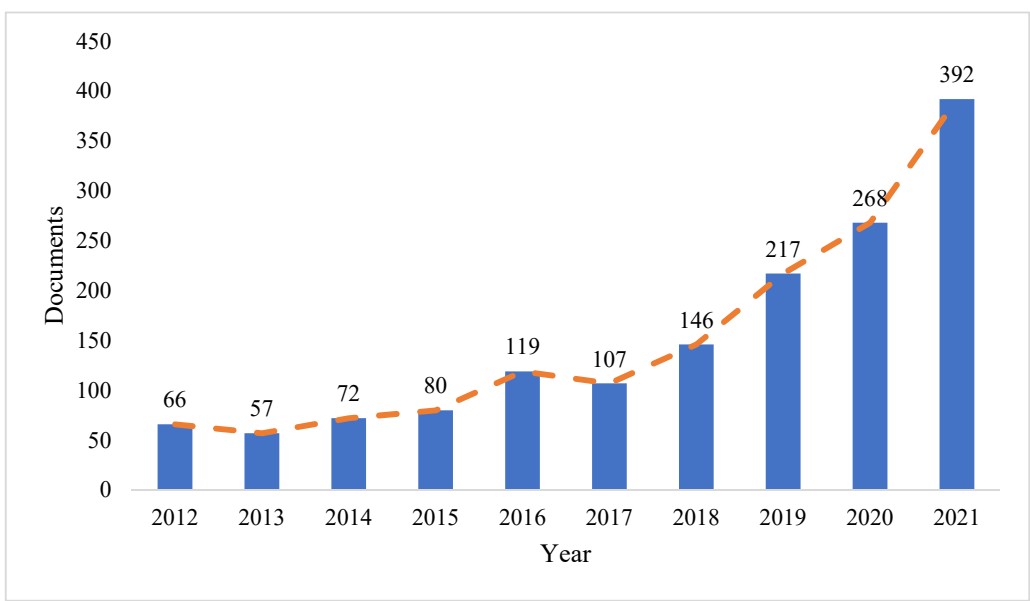

**Figure 1.** Growth of publications in the last 10 years.

As shown in Table 1, these documents belong in greater proportion to the areas of business, management and accounting, social sciences, computer science, environmental science, and engineering. Other subject areas such as energy, economics, econometrics and finance, and decision sciences are also highlighted. It should be clarified that the sum of the "Documents" column exceeds the number of documents published in 2012–2021, as some documents and journals belong to several subject areas. In these subject areas, the approach to the main dimensions of sustainability (economic, social, and environmental) is evidenced; they are fields that directly address open innovation, such as business, management and accounting, and social sciences, or fields directly related to e-marketplaces such as computer science, economics, econometrics and finance, and decision sciences.

**Table 1.** Subject areas for 2012–2021.

| Subject Area | Documents | % Ind. | % Accum. |
|:---:|:---:|:---:|:---:|
| Business, Management and Accounting | 517 | 15.8% | 15.8% |
| Social Sciences | 512 | 15.6% | 31.4% |
| Computer Science | 419 | 12.8% | 44.2% |
| Environmental Science | 372 | 11.4% | 55.6% |
| Engineering | 338 | 10.3% | 65.9% |
| Energy | 326 | 10.0% | 75.9% |
| Economics, Econometrics and Finance | 201 | 6.1% | 82.0% |
| Decision Sciences | 176 | 5.4% | 87.4% |
| Others | 375 | 12,5% | 100% |

Table 2 shows the publications in which the most documents are disclosed (the leading journals); they contribute 29.3% of the documents in the last ten years. These journals include Sustainability Switzerland, Journal of Open Innovation Technology Market and Complexity, Journal of Cleaner Production, and Advances in Intelligent Systems and Computing, with a contribution of 25 or more documents. The scope of these journals, book series and conference proceedings is related to environmental, cultural, economic, and social sustainability, open innovation, open business models, cleaner production and environmental research, methods of intelligent systems, and computing. Likewise, more

than half of the publications in Table 2 correspond to conference proceedings and book series, while the journals belong to quartiles Q1 and Q2 in Scimago.

**Table 2.** Leading journals for 2012–2021.

| Publication | Docs. | % Docs. | Publication Type | *h*-Index 2020 (Scimago) | Max Quartil 2020 (Scimago) |
|---|---|---|---|---|---|
| Sustainability Switzerland | 195 | 12.8% | Journals | 85 | Q1 |
| Journal of Open Innovation Technology Market and Complexity | 59 | 3.9% | Journals | 22 | Q2 |
| Journal of Cleaner Production | 30 | 2.0% | Journals | 200 | Q1 |
| Advances in Intelligent Systems and Computing | 25 | 1.6% | Book Series | 41 | N/A |
| E3s Web of Conferences | 19 | 1.2% | Conferences and Proceedings | 22 | N/A |
| Lecture Notes in Computer Science | 19 | 1.2% | Book Series | 400 | Q3 |
| IOP Conference Series Earth and Environmental Science | 17 | 1.1% | Conferences and Proceedings | 26 | N/A |
| ACM International Conference Proceeding Series | 16 | 1.0% | Conferences and Proceedings | 123 | N/A |
| IFIP Advances in Information and Communication Technology | 12 | 0.8% | Book Series | 53 | Q3 |
| Smart Innovation Systems and Technologies | 9 | 0.6% | Book Series | 22 | Q4 |
| Communications in Computer and Information Science | 8 | 0.5% | Book Series | 51 | Q4 |
| IOP Conference Series Materials Science and Engineering | 8 | 0.5% | Conferences and Proceedings | 44 | N/A |
| Journal Of Physics Conference Series | 8 | 0.5% | Conferences and Proceedings | 85 | Q4 |

Regarding the authors who published the most documents in the last ten years (the leading authors), Table 3 highlights Park, Ramírez-Montoya, Callou, Saguy, and Yun, who have published seven or more documents in the last ten years. Park is the one of the lead authors, and he publishes in co-authorship with Yun on issues of sustainability and open innovation, along with their impact on business models for different industries [28,35,60]. Ramírez-Montoya is the other lead author, focusing on education and knowledge innovation towards sustainable development [61–63].

**Table 3.** Leading authors for 2012–2021.

| Author | Docs. * | Scopus Author ID | *h*-Index and Citations | Main Subject Area | Affiliation | Country |
|---|---|---|---|---|---|---|
| Park, K.B. | 8 | 56828377000 | *h*-index:18 967 citations by 672 documents | Absorptive Capacity; Open Innovation; Business Model Innovation; Sustainable Business; Digital Transformation | Sangji University | Wonju, South Korea |
| Ramírez-Montoya, M.S. | 8 | 54911980200 | *h*-index: 14 622 citations by 455 documents | Online Courses; Learner Behaviour; Blended Learning; Learning Environment; Educational Innovation | Tecnologico de Monterrey | Monterrey, Mexico |
| Callou, G. | 7 | 25633749100 | *h*-index: 13 543 citations by 408 documents | Rejuvenation; Stochastic Petri Nets; Continuous-Time Markov Chain | Universidade Federal Rural de Pernambuco | Recife, Brazil |
| Saguy, I.S. | 7 | 7003669734 | *h*-index: 37 4266 citations by 3403 documents | Open Innovation; Food Sector; Product and Process Innovation; Education for Sustainability; Higher Education Institutions; Sustainability Science and Engineering | Department of Food Science and Nutrition | Jerusalem, Israel |
| Yun, J.H.J. | 7 | 55419994900 | *h*-index: 21 1725 citations by 938 documents | Open Innovation; Absorptive Capacity; Business Model Innovation; Sustainable Business; Digital Transformation | Daegu Gyeongbuk Institute of Science and Technology | Daegu, South Korea |
| Costa, J. | 6 | 57212821686 | *h*-index: 5 74 citations by 68 documents | Open Innovation; Alliance Portfolios; Absorptive Capacity; Community Innovation Survey; Marketing Innovation; Manufacturing Firms | Universidade de Aveiro | Aveiro, Portugal |
| Gatta, V. | 6 | 35109007100 | *h*-index: 24 1191 citations by 558 documents | Urban Freight Transport; City Logistics; Cargo | Università degli Studi Roma Tre | Rome, Italy |
| Mangiaracina, R. | 6 | 35325033800 | *h*-index: 14 762 citations by 657 documents | Electronic Commerce; Groceries; Logit Equilibrium; Logistics Service Providers; 3Pl; Third-party Logistics | Politecnico di Milano | Milan, Italy |
| Marcucci, E. | 6 | 6602255083 | *h*-index: 28 1912 citations by 1068 documents | Urban Freight Transport; City Logistics; Cargo | Høgskolen i Molde | Molde, Norway |
| Tumino, A. | 7 | 25929706400 | *h*-index: 13 771 citations by 664 documents | Electronic Commerce; Groceries; Logit Equilibrium; Logistics Service Providers; 3Pl; Third-party Logistics | Politecnico di Milano | Milan, Italy |
| Tutsch, D. | 6 | 6506708815 | *h*-index: 8 256 citations by 190 documents | Petri Nets; Concurrent Systems; Programmable Logic Controllers; Rejuvenation; Continuous-Time Markov Chain | Bergische Universität Wuppertal | Wuppertal, Germany |

* Documents published in Scopus in 2012–2021 related to open innovation, sustainability, and e-marketplaces.

Callou and Tutsch usually co-author on issues of sustainability and the environmental impact of data center systems [64–66]. Saguy focuses on the study of open innovation in SMEs in the food industry [67,68]. Costa researches on open innovation for sustainable innovation ecosystems [36] and open innovation for university-industry linkage [69]. Gatta and Marcucci publish in co-authorship on sustainable urban freight transport and urban mobility [70–72]. Mangiaracina and Tumino usually publish in co-authorship on innovative solutions and the environmental assessment of logistics in B2C e-commerce [73–75]. The scientific output by country/territory is shown in Figure 2, recognizing China as the leader with 233 documents (11.0%), followed by the United States with 128 documents (6.0%), the United Kingdom (4.7%), Italy (4.5%), India (4.5%), and Germany (4.4%). Likewise, the European leading countries contribute 29.2% of the total documents, the Asian leading countries contribute 22.8%, Australia leads in Oceania with 2.2% of the documents, and Brazil leads in South America (1.9%).

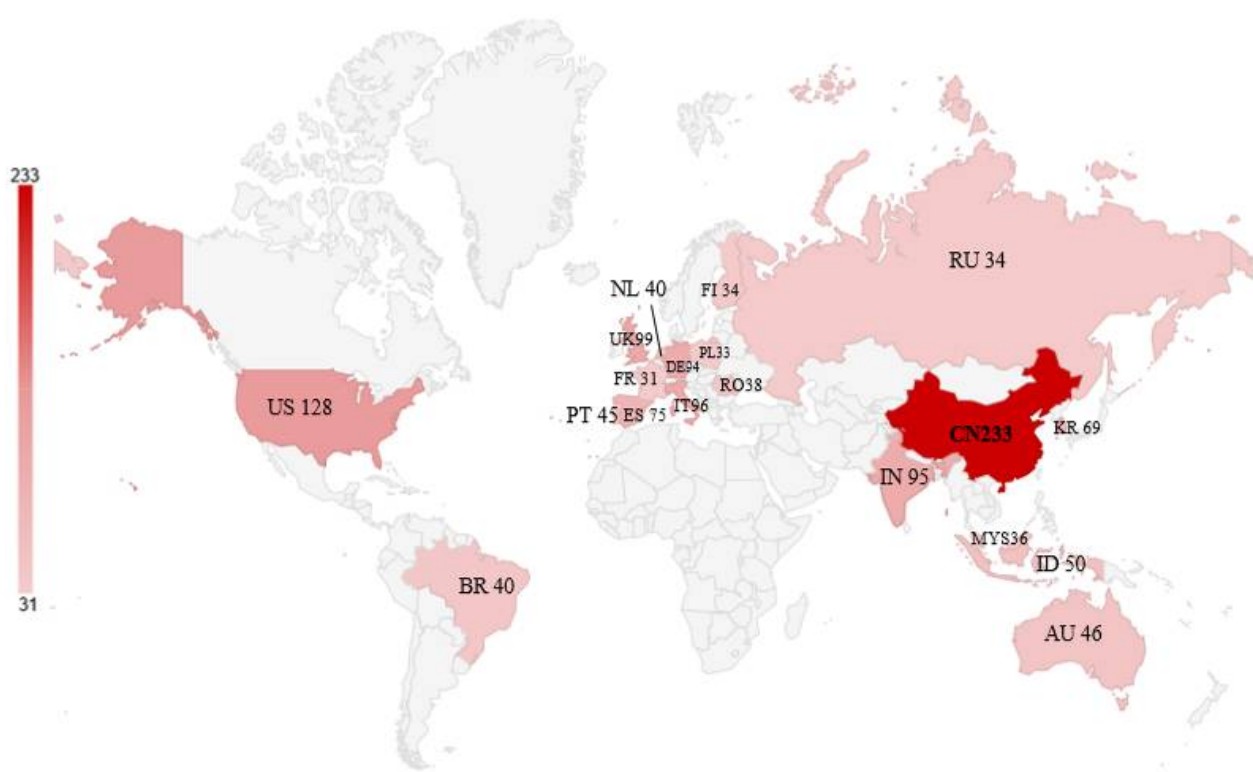

**Figure 2.** Scientific output by country/territory in 2012–2021.

For the long-term analysis, Table 4 shows that the leading affiliation with the most documents published is the Tecnologico de Monterrey (Mexico), with 16 documents (2.0%). This institution is followed by the Politecnico di Milano (Italy), Universidade de Aveiro (Portugal), Bucharest University of Economic Studies (Romania), Universidad Rey Juan Carlos (Spain), Delft University of Technology (Netherlands), Università degli Studi Roma Tre (Italy), and Daegu Gyeongbuk Institute of Science and Technology (South Korea). Except for Tecnologico de Monterrey, the other leading affiliations correspond to European, Korean, and Australian universities, which individually contributed more than seven documents in the last ten years.

**Table 4.** Leading affiliations for 2012–2021.

| Affiliation | Documents | % Ind. | % Accum. |
| --- | --- | --- | --- |
| Tecnologico de Monterrey | 16 | 2.0% | 2.0% |
| Politecnico di Milano | 13 | 1.7% | 3.7% |
| Universidade de Aveiro | 13 | 1.7% | 5.3% |
| Bucharest University of Economic Studies | 13 | 1.7% | 7.0% |
| Universidad Rey Juan Carlos | 12 | 1.5% | 8.5% |
| Delft University of Technology | 10 | 1.3% | 9.8% |
| Università degli Studi Roma Tre | 10 | 1.3% | 11.1% |
| Daegu Gyeongbuk Institute of Science and Technology | 10 | 1.3% | 12.3% |
| Wageningen University & Research | 9 | 1.1% | 13.5% |
| Queensland University of Technology | 9 | 1.1% | 14.6% |
| Technical University of Berlin | 9 | 1.1% | 15.8% |
| Sangji University | 9 | 1.1% | 16.9% |
| Hebrew University of Jerusalem | 8 | 1.0% | 17.9% |
| Aristotle University of Thessaloniki | 8 | 1.0% | 18.9% |
| Luiss University | 8 | 1.0% | 19.9% |
| Universidad Politécnica de Madrid | 8 | 1.0% | 21.0% |
| Bina Nusantara University | 8 | 1.0% | 22.0% |

The number of citations per document is one of the ways to measure the impact of studies generated around e-marketplaces, open innovation, and sustainability. Therefore, Table 5 describes the documents with the highest impact in the last ten years, the main topics addressed, and the number of citations in Scopus. Among the most cited documents, we identified that four of them belong to the Journal of Cleaner Production, two of them belong to Sustainability (Switzerland), and two of them belong to Technological Forecasting and Social Change; these journals belong to the leading journals from Table 2. Additionally, two articles in Table 5 belong to the Journal of Product Innovation Management. On the other hand, the documents receive an average of between 7.1 and 38.3 citations per year.

**Table 5.** Most cited documents for 2012–2021.

| Document Title | Topics | Authors | Cites |
| --- | --- | --- | --- |
| Towards an effective framework for building smart cities: Lessons from Seoul and San Francisco Change | The authors study the process of building an effective smart city by integrating various practical perspectives based on the literature, especially for the case of San Francisco and Seoul Metropolitan City. The coordination of activities and resources on an open innovation platform allowed the coordination between public and private sector actors to enable effective, sustainable smart cities. They analyze urban openness, service information, partnership formation, urban proactiveness, smart city infrastructure integration, and smart city governance. | [76] | 416 |
| The 1% rule in four digital health social networks: An observational study | The research is focused on the 1% rule, or 90-9-1 principle. This rule state that 90% of actors observe and do not participate (Lurkers), 9% contribute sparingly (Contributors), and 1% of actors create most of the new content (Superusers) within Internet communities. Authors apply the 1% rule to moderated Digital Health Social Networks, highlighting that Superusers generate most of the traffic and create value, so their recruitment and retention is imperative for long-term success. | [77] | 181 |

**Table 5.** *Cont.*

| Document Title | Topics | Authors | Cites |
|---|---|---|---|
| Changing R&D models in research-based pharmaceutical companies | The study aims to identify, analyze, and describe the factors that impact the R&D efficiency of major research-based pharmaceutical companies and analyzed the key challenges and success factors of a sustainable R&D output. Authors analyzed the concepts that companies follow to increase their R&D efficiencies: Activities to reduce portfolio and project risk, activities to reduce R&D costs, and activities to increase the innovation potential. Authors suggest following some open innovators such as knowledge creator, knowledge integrator or knowledge leverage. | [78] | 157 |
| Micro- and macro-dynamics of open innovation with a Quadruple-Helix model | This paper explores how sustainability can be achieved through open innovation in the current 4th industrial revolution. The authors identify micro and macro dynamics of open innovation, the dynamic roles of industry, government, university, and society, and propose a conceptual framework to understand open innovation dynamics with a quadruple-helix model for social, environmental, economic, cultural, policy, and knowledge sustainability. | [54] | 139 |
| Product development and management association's 2012 comparative performance assessment study | Authors present the results of a comparative performance assessment study for product development (PMDA) that introduces new sections on culture, social media, services, sustainability, open innovation, and global product development practices to reveal practices that lead to higher product performance in the market. | [79] | 134 |
| Sustainable business models: A review | This research provides a comprehensive review of sustainable business models literature in various application areas, and classifies notable sustainable business models according to innovation, management and marketing, entrepreneurship, energy, fashion, healthcare, agri-food, supply chain management, circular economy, developing countries, engineering, construction and real estate, mobility and transportation, and hospitality. | [27] | 122 |
| An analysis of the interplay between organizational sustainability, knowledge management, and open innovation | This paper explores the case of a Brazilian family-owned company of rubber products, operating in the sectors of health, education, and coatings. This company uses knowledge to develop open innovation aiming to promote sustainable innovation since open innovation plays a key role towards effective strategic sustainable management. Authors determine that knowledge management and open innovation promotes sustainable innovations. | [31] | 118 |
| User-integrated innovation in Sustainable LivingLabs: An experimental infrastructure for researching and developing sustainable product service systems | This study presents the Sustainable LivingLabs (SLL) research infrastructure and its methodology (insight research, prototyping, field testing) involving real-life socio-technical experiments and the implementation of sustainable product service systems (PSS). | [80] | 114 |
| Technological challenges of green innovation and sustainable resource management with large scale data | This paper presents an overview of articles about sustainable development papers based on big data, the relationship between environmental pollution and influencing factors, and sustainable natural resource management based on large scale data. Authors highlight that many additional challenges must be solved to establish and support systems which will guide and monitor transformations into sustainable, livable, and low pollution. | [81] | 106 |
| Designing coupled innovations for the sustainability transition of agrifood systems | This paper provides a framework to organize the design of coupled innovations, by reconnecting the dynamics of innovation (technological, organizational, and institutional innovations) in agriculture and food industries to improving the sustainability in the whole agri-food system. Authors conclude that the need for innovation in agri-food systems requires going beyond the specialization of skills, and the usual forms of coordination between designers. | [82] | 102 |

**Table 5.** *Cont.*

| Document Title | Topics | Authors | Cites |
|---|---|---|---|
| Orchestrating' sustainable crowdsourcing: A characterisation of solver brokerages | Authors examines the 'Solver Brokerage,' which enables innovation exchanges between organizations and unknown external firms and individuals (crowdsourcing process). They examine research on innovation networks, crowdsourcing, and electronic marketplaces to identify three knowledge mobility, appropriability and stability processes that are necessary to orchestrate crowdsourcing. | [83] | 100 |
| The emerging research landscape on bioeconomy: What has been done so far and what is essential from a technology and innovation management perspective? | Authors conduct an overview of the current research landscape dealing with the bioeconomy. The study reveals that the evolution of the bioeconomy is still on a strategic level and open innovation enables a holistic view on organizing future resource allocation and biomass flow across value chains. It suggests that essential innovation management related research frames might contribute to a sustainable evolution of the bioeconomy by addressing the major challenges. | [84] | 96 |
| Living labs: Implementing open innovation in the public sector | The research contributes to understand the role of living labs as intermediaries of public open innovation. Authors analyze two living labs: Citilab in the city of Cornellà) and public fab labs in the city of Barcelona. Among the conclusions, the study highlights that scalability and sustainability are the main problems living labs encounter as open innovation intermediaries. | [85] | 95 |
| Open innovation and its effects on economic and sustainability innovation performance | The authors investigate the roles that different open innovation partners played in improving economic innovation performance and sustainability innovation performance. Authors found that economic innovation performance positively correlates with sustainability innovation performance, which implies that economic and sustainability innovation goals can be reached simultaneously. | [46] | 94 |
| Sustainable business models and structures for industry 4.0 | The paper addresses the research question of how new and sustainable business models and structures for Industry 4.0 might look like and in which direction existing traditional business concepts must be developed to deploy a strong business impact of Industry 4.0. | [56] | 91 |
| A review of the environmental implications of B2C e-commerce: a logistics perspective | This study offers a literature review on the topic of B2C e-commerce environmental sustainability, specifically from a logistics perspective, highlighting the need for a quantitative evaluation of environmental impact of e-commerce. | [25] | 87 |
| The front-end of eco-innovation for eco-innovative small and medium sized companies | This paper investigates the Front End of Eco-Innovation (FEEI)—the initial stages of the eco-innovation process—for 42 small and medium sized eco-innovators in the Netherlands. The results show that SMEs embrace informal, systematic, and open innovation approaches at the FEEI. | [86] | 85 |
| Carbon emissions in a dual channel closed loop supply chain: the impact of consumer free riding behavior | The research evaluates the impact of consumer free riding on carbon emissions in a product's life cycle across a dual channel closed loop supply chain (traditional retailers and online e-tailers) and assesses the effect of governmental e-commerce tax on carbon emissions. | [87] | 82 |
| Harnessing Difference: A Capability-Based Framework for Stakeholder Engagement in Environmental Innovation | Authors present a systematic review to enhance understanding of how firms can effectively incorporate stakeholder perspectives for environmental innovation. The study shows that engaging stakeholders in environmental innovation requires specific operational capabilities, engagement management capabilities, and capabilities to co-create innovative solutions and to learn from stakeholder engagement activities (systematized learning). | [88] | 80 |
| A comparative analysis of carbon emissions from online retailing of fast moving consumer goods | This study develops a framework considering all the relevant environmental factors relating to retail/e-commerce activities to build a Life Cycle Analysis model. Variables such as basket size, transport mode, trip length, trip frequency, the amount and type of packaging used, and the energy efficiency of e-fulfilment operations are included in the analysis to determine the environmental sustainability of e-commerce. | [49] | 78 |

The main topics addressed among the most cited documents in 2012–2021 are related to e-commerce and environmental sustainability [25,49,87], sustainability achieved through open innovation [31,46,54,88], sustainable business models [27,49], and open innovation and living labs [80,85]. Likewise, the topics addressed in this period are related to open innovation and bioeconomy [84], open innovation and smart cities [76], open innovation and R&D efficiency [78], crowdsourcing [83], eco-innovation [86], digital social networks [77], product success and product development [79], and innovation in the agriculture and food industries [82]. Among the main research concepts addressed between 2012 and 2021, Figure 3 shows that sustainable development, open innovation, electronic commerce, sustainability, innovation, and e-commerce stand out. As for minor themes, the concepts of competition, sales, marketing, information management, and planning were identified.

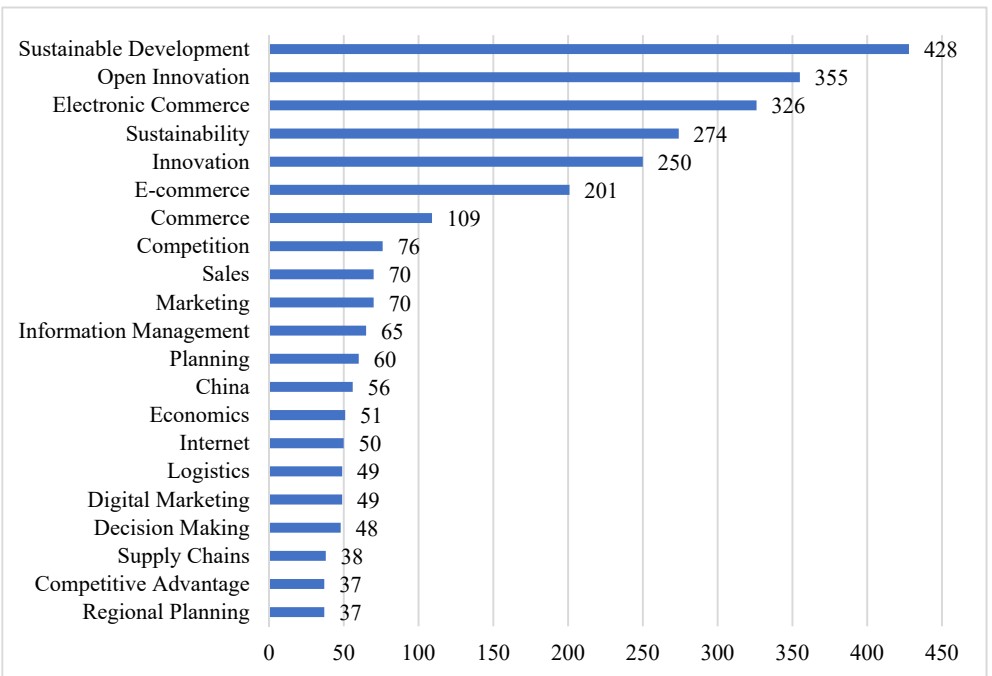

**Figure 3.** Main concepts in 2012–2021.

The association of these concepts is detailed in Figure 4, which identifies clusters around the main topics. These clusters are identified using the item filter and by zooming in in the Vosviewer software in order to associate the colors of the nodes to each concept. The central node of each cluster is identified through the sphere size, font size, and the number of connections with other nodes. In Cluster 1 (the red cluster), sustainability is identified as the central node, around which topics such as marketing, China, logistics, the Internet, and supply chain management are clustered. In Cluster 2 (the green cluster), the main node is sustainable development, grouping terms which are different from those in Figure 3, such as environmental impact, environmental sustainability, city and urban logistics, traffic congestion, and packaging. The central node of Cluster 3 (the blue cluster) is open innovation, relating topics of digital marketing, knowledge management, information systems, business models, smart cities, and co-creation. Innovation represents the central node of Cluster 4 (the yellow cluster), grouping concepts such as product development, industry, and collaboration. Cluster 5 (the purple cluster) focuses on electronic commerce and e-commerce, surrounded by topics of competition, sales, decision making, competitive advantages, and online shopping. Cluster 6 (the light-blue cluster) focuses on commerce, grouping topics of information management, planning, economics, regional planning, and supply chains.

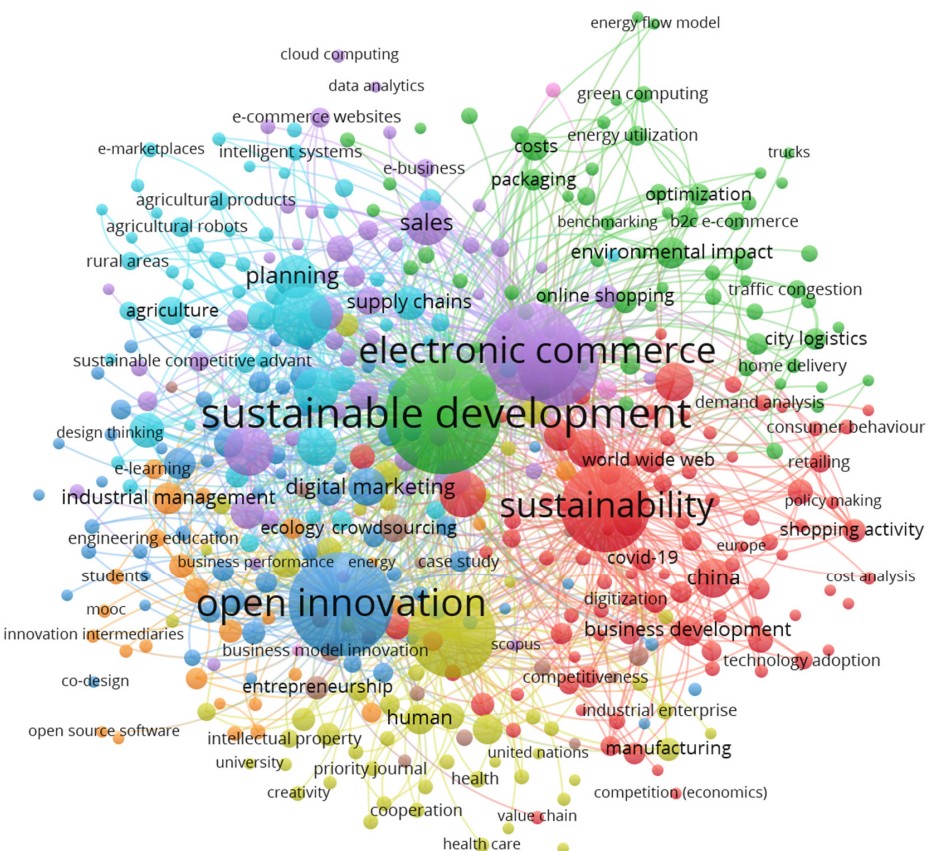

**Figure 4.** Co-occurrence analysis in 2012–2021.

### 3.2. Medium-Term Bibliometric Analysis: 2017–2021

For the analysis of the last five years, 1130 documents were obtained from Scopus, representing 74.1% of the published documents in the last ten years. This means that the topics of electronic markets, open innovation, and sustainability have been developed mostly in recent years, representing nascent and young topics in the literature. Regarding the subject areas in the medium term, Table 6 shows that social sciences have become the most relevant area, after being in the second position in the long term. Environmental science rises to the third position after being in the fourth position, while computer science falls to the fifth position after being in the third position. In this way, changes in the research fields are identified, with social, business, and environmental issues becoming more relevant in the medium term.

**Table 6.** Most cited documents for 2017–2021.

| Subject Area | Document | % Ind. | % Accum. |
|---|---|---|---|
| Social Sciences | 440 | 17.2% | 17.2% |
| Business, Management and Accounting | 354 | 13.8% | 31.0% |
| Environmental Science | 330 | 12.9% | 43.9% |
| Energy | 296 | 11.6% | 55.4% |
| Computer Science | 281 | 11.0% | 66.4% |
| Engineering | 250 | 9.8% | 76.2% |
| Economics, Econometrics and Finance | 159 | 6.2% | 82.4% |
| Decision Sciences | 127 | 5.0% | 87.3% |
| Others | 325 | 12.6% | 100% |

The most representative journals of the last five years, with the most publications, also belong to the most relevant journals of the last ten years. Sustainability Switzerland, Journal of Open Innovation Technology Market and Complexity, Journal of Cleaner Production, Advances in Intelligent Systems and Computing, and E3s Web of Conferences remain in the same positions. The leading journals from Table 7 account for 33.3% of the total publications in the medium term. As a novelty, IOP Conference Series Earth and Environmental Science rise from seventh to sixth place, and only Communications in Computer and Information Science disappears from this list.

**Table 7.** Leading journals for 2017–2021.

| Publication | Docs. | % Docs. | Publication Type | *h*-Index 2020 (Scimago) | Max Quartil 2020 (Scimago) |
|---|---|---|---|---|---|
| Sustainability Switzerland | 185 | 16.4% | Journals | 85 | Q1 |
| Journal of Open Innovation Technology Market and Complexity | 57 | 5.0% | Journals | 22 | Q2 |
| Journal of Cleaner Production | 25 | 2.2% | Journals | 200 | Q1 |
| Advances In Intelligent Systems and Computing | 20 | 1.8% | Book Series | 41 | N/A |
| E3s Web of Conferences | 19 | 1.7% | Conferences and Proceedings | 22 | N/A |
| IOP Conference Series Earth and Environmental Science | 17 | 1.5% | Conferences and Proceedings | 26 | N/A |
| Lecture Notes in Computer Science | 12 | 1.1% | Book Series | 400 | Q3 |
| ACM International Conference Proceeding Series | 9 | 0.8% | Conferences and Proceedings | 123 | N/A |
| Smart Innovation Systems and Technologies | 9 | 0.8% | Book Series | 22 | Q4 |
| IFIP Advances in Information and Communication Technology | 8 | 0.7% | Book Series | 53 | Q3 |
| Journal Of Physics Conference Series | 8 | 0.7% | Conferences and Proceedings | 85 | Q4 |
| IOP Conference Series Materials Science and Engineering | 7 | 0.6% | Conferences and Proceedings | 44 | N/A |

Table 8 details the leading authors for 2017–2021, of which Costa, Gatta, Marcucci, Park, Ramírez-Montoya, Callou, Mangiaracina, Tumino, and Yun remain in the ranking of the most cited authors for the medium and long terms, and Park remains the most cited author in the long and medium terms. Likewise, authors such as Abreu, Buldeo Rai, Carayannis, Liu, Macharis, Zhao, Wang, and Dang appear in the ranking of leading authors. Abreu addresses issues of innovation ecosystems from a sustainability perspective [89–91]. Buldeo Rai and Macharis co-author studies related to sustainability, delivery, and e-commerce delivery [92–94]. Wang co-authored with Dang on sustainable supply chains and e-commerce logistics [95,96]. Carayannis addresses issues related to business model innovation and sustainability [97–99]. Liu performs studies in which sustainability is related to innovation ecosystems, technology innovation, and open innovation [100–102]. Zhao co-authors with Park and Yun, mainly on topics of Sustainability and Open Innovation [28,35].

**Table 8.** Leading authors for 2017–2021.

| Author | Docs. * | Scopus Author ID | *h*-Index and Citations | Main Subject Area | Affiliation | Country |
|---|---|---|---|---|---|---|
| Costa, J. | 6 | 57212821686 | *h*-index: 5 74 citations by 68 documents | Open Innovation; Alliance Portfolios; Absorptive Capacity; Community Innovation Survey; Marketing Innovation; Manufacturing Firms | Universidade de Aveiro | Aveiro, Portugal |
| Gatta, V. | 6 | 35109007100 | *h*-index: 24 1191 citations by 558 documents | Urban Freight Transport; City Logistics; Cargo | Università degli Studi Roma Tre | Rome, Italy |
| Marcucci, E. | 6 | 6602255083 | *h*-index: 28 1912 citations by 1068 documents | Urban Freight Transport; City Logistics; Cargo | Høgskolen i Molde | Molde, Norway |
| Park, K.B. | 6 | 56828377000 | *h*-index:18 967 citations by 672 documents | Absorptive Capacity; Open Innovation; Business Model Innovation; Sustainable Business; Digital Transformation | Sangji University | Wonju, South Korea |
| Ramírez-Montoya, M.S. | 5 | 54911980200 | *h*-index: 14 622 citations by 455 documents | Online Courses; Learner Behaviour; Blended Learning; Learning Environment; Educational Innovation | Tecnologico de Monterrey | Monterrey, Mexico |
| Abreu, A. | 4 | 57218315486 | *h*-index: 12 380 citations by 252 documents | Zero Energy Buildings; Refurbishment; Renovation; Alliance Portfolios; Absorptive Capacity; Open Innovation | Instituto Superior de Engenharia de Lisboa | Lisbon, Portugal |
| Buldeo Rai, H. | 4 | 57195135422 | *h*-index: 12 260 citations by 217 documents | Urban Freight Transport; City Logistics; Cargo | Vrije Universiteit Brussel | Brussels, Belgium |
| Callou, G. | 4 | 25633749100 | *h*-index: 13 543 citations by 408 documents | Rejuvenation; Stochastic Petri Nets; Continuous-Time Markov Chain | Universidade Federal Rural de Pernambuco | Recife, Brazil |
| Carayannis, E.G. | 4 | 7006225155 | *h*-index: 38 5364 citations by 3858 documents | Quadruple; Triple Helix; Artistic Research; Alliance Portfolios; Absorptive Capacity; Open Innovation | GW School of Business | Washington, D.C., United States |
| Dang, T.T. | 4 | 57218565464 | *h*-index: 6 113 citations by 73 documents | Decision-making, e-commerce marketplaces, Logistics, Last Mile Delivery | International University, Vietnam National University Ho Chi Minh City | Ho Chi Minh City, Viet Nam |
| Liu, Z. | 4 | 57190680108 | *h*-index: 4 179 citations by 164 documents | Alliance Portfolios; Absorptive Capacity; Open Innovation | Cardiff School of Management | Cardiff, United Kingdom |
| Macharis, C. | 4 | 6507193118 | *h*-index: 39 5524 citations by 3944 documents | Urban Freight Transport; City Logistics; Cargo | Vrije Universiteit Brussel | Brussels, Belgium |
| Mangiaracina, R. | 4 | 35325033800 | *h*-index: 14 762 citations by 657 documents | Electronic Commerce; Groceries; Logit Equilibrium; Logistics Service Providers; 3Pl; Third-party Logistics | Politecnico di Milano | Milan, Italy |

**Table 8.** *Cont.*

| Author | Docs. * | Scopus Author ID | *h*-Index and Citations | Main Subject Area | Affiliation | Country |
|--------|---------|------------------|--------------------------|-------------------|-------------|---------|
| Tumino, A. | 4 | 25929706400 | *h*-index: 13 771 citations by 664 documents | Electronic Commerce; Groceries; Logit Equilibrium; Logistics Service Providers; 3Pl; Third-party Logistics | Politecnico di Milano | Milan, Italy |
| Wang, C.N. | 4 | 7501640993 | *h*-index: 17 1170 citations by 884 documents | Decision-making, e-commerce marketplaces, Logistics, Last Mile Delivery | National Kaohsiung University of Science and Technology | Kaohsiung, Taiwan |
| Yun, J.H.J. | 4 | 55419994900 | *h*-index: 21 1725 citations by 938 documents | Open Innovation; Absorptive Capacity; Business Model Innovation; Sustainable Business; Digital Transformation | Daegu Gyeongbuk Institute of Science and Technology | Daegu, South Korea |
| Zhao, X. | 4 | 57193208482 | *h*-index: 14 567 citations by 419 documents | Open Innovation; Absorptive Capacity; Business Model Innovation; Sustainable Business; Digital Transformation | Daegu Gyeongbuk Institute of Science and Technology | Daegu, South Korea |

* Documents published in Scopus in 2017–2021 related to Open Innovation, Sustainability, and e-marketplaces.

As shown in Figure 5, 20 countries contributed more than 20 documents in the medium term, making up 68.3% of documents from the last five years. China and the United States remain leaders in publications in the medium term, with 189 and 88 documents, respectively. In the medium term, India increases two positions to occupy the third place, and Italy continues to occupy the fourth place, while the United Kingdom goes from the third place to the fifth place. On the other hand, Greece enters the ranking with 23 documents in the last five years, and the leading countries from Europe contribute 27.1% of the documents, as well as the leading countries from Asia, contributing 26.5% of documents.

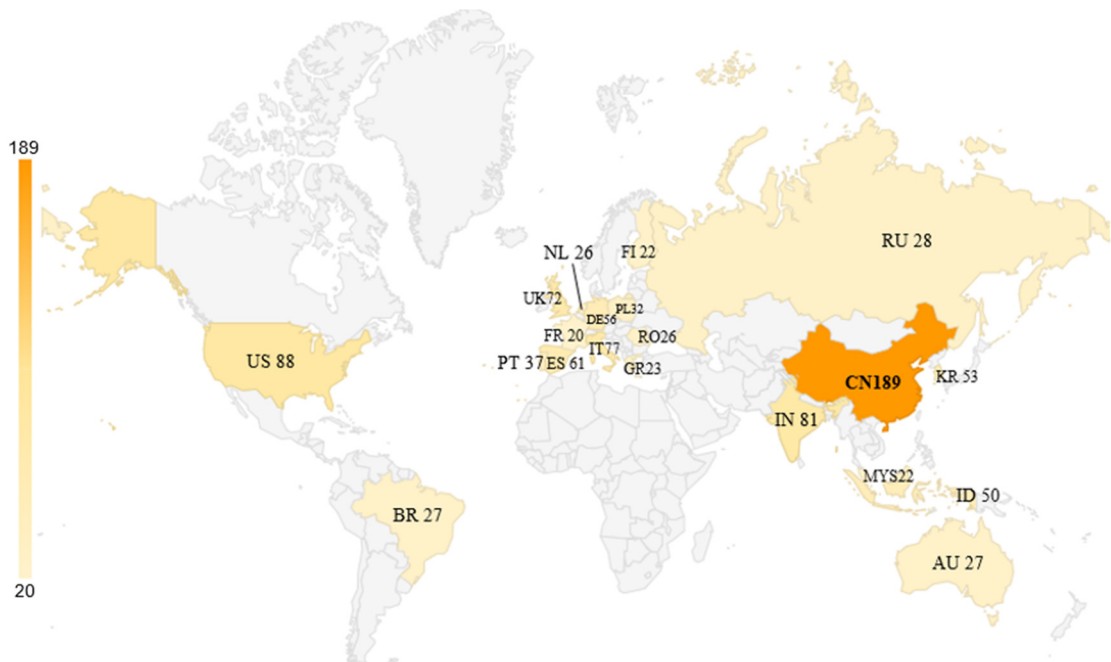

**Figure 5.** Scientific output by country/territory in 2017–2021.

Moreover, Table 9 describes the leading affiliations, highlighting that Universidade de Aveiro ranks first in the medium term (it publishes an average of 2.6 documents per year) after ranking third in the long term. Tecnologico de Monterrey maintains a good position, going from first to fourth place in that period, and Universidad Rey Juan Carlos rises from fifth to the second position. Unlike the leading affiliations of the last ten years, in this case, Peter the Great St. Petersburg Polytechnic University and Università degli Studi di Napoli Federico II appear, and Delft University of Technology, Wageningen University and Research, Sangji University, Hebrew University of Jerusalem, Aristotle University of Thessaloniki, and Luiss University disappear. On average, the leading affiliations contribute 1.6 documents per year.

**Table 9.** Leading affiliations for 2017–2021.

| Affiliation | Documents | % Ind. | % Accum. |
|---|---|---|---|
| Universidade de Aveiro | 13 | 2.0% | 2.0% |
| Universidad Rey Juan Carlos | 11 | 1.7% | 3.7% |
| Politecnico di Milano | 11 | 1.7% | 5.4% |
| Tecnologico de Monterrey | 9 | 1.4% | 6.8% |
| Università degli Studi Roma Tre | 8 | 1.2% | 8.0% |
| Bina Nusantara University | 8 | 1.2% | 9.2% |
| Queensland University of Technology | 7 | 1.1% | 10.3% |
| Technical University of Berlin | 7 | 1.1% | 11.4% |
| Peter the Great St. Petersburg Polytechnic University | 7 | 1.1% | 12.4% |
| Università degli Studi di Napoli Federico II | 7 | 1.1% | 13.5% |
| Universidad Politécnica de Madrid | 7 | 1.1% | 14.6% |
| Daegu Gyeongbuk Institute of Science and Technology | 7 | 1.1% | 15.7% |
| Bucharest University of Economic Studies | 7 | 1.1% | 16.7% |

Table 10 indicates that the three most cited documents in the medium term are also part of the ranking of the most cited documents in the long term, which demonstrates their high impact through the receipt of an average of 46 citations per year [54], 40 citations per year [31], and 23 citations per year [27]. The annual citation rate for the documents in Table 10 is 15.1 citations per year, highlighting that 25% of the most cited documents belong to the Journal of Cleaner Production, which presents an average of 90 citations in five years and 18 citations per year. Likewise, 25% of the most cited documents belong to Sustainability (Switzerland), with an average of 55 citations in five years and 11 citations per year.

**Table 10.** Most cited documents in 2017–2021.

| Document Title | Topics | Authors | Cites |
|---|---|---|---|
| Micro- and macro-dynamics of open innovation with a Quadruple-Helix model | This paper explores how sustainability can be achieved through open innovation in the current 4th industrial revolution. The authors identify micro and macro dynamics of open innovation, the dynamic roles of industry, government, university, and society, and propose a conceptual framework to understand open innovation dynamics with a quadruple-helix model for social, environmental, economic, cultural, policy, and knowledge sustainability. | [54] | 139 |
| Sustainable business models: A review | This research provides a comprehensive review of sustainable business models literature in various application areas, and classifies notable sustainable business models according to innovation, management and marketing, entrepreneurship, energy, fashion, healthcare, agri-food, supply chain management, circular economy, developing countries, engineering, construction and real estate, mobility and transportation, and hospitality. | [27] | 122 |

**Table 10.** *Cont.*

| Document Title | Topics | Authors | Cites |
|---|---|---|---|
| An analysis of the interplay between organizational sustainability, knowledge management, and open innovation | This paper explores the case of a Brazilian family-owned company of rubber products, operating in the sectors of health, education, and coatings. This company uses knowledge to develop open innovation aiming to promote sustainable innovation since open innovation plays a key role towards effective strategic sustainable management. Authors determine that knowledge management and open innovation promotes sustainable innovations. | [31] | 118 |
| Technological challenges of green innovation and sustainable resource management with large scale data | This paper presents an overview of articles about sustainable development papers based on big data, the relationship between environmental pollution and influencing factors, and sustainable natural resource management based on large scale data. Authors highlight that many additional challenges must be solved to establish and support systems which will guide and monitor transformations into sustainable, livable, and low pollution. | [81] | 106 |
| Designing coupled innovations for the sustainability transition of agrifood systems | This paper provides a framework to organize the design of coupled innovations, by reconnecting the dynamics of innovation (technological, organizational, and institutional innovations) in agriculture and food industries to improving the sustainability in the whole agri-food system. Authors conclude that the need for innovation in agri-food systems requires going beyond the specialization of skills, and the usual forms of coordination between designers. | [82] | 102 |
| Living labs: Implementing open innovation in the public sector | The research contributes to understand the role of living labs as intermediaries of public open innovation. Authors analyze two living labs: Citilab in the city of Cornellà) and public fab labs in the city of Barcelona. Among the conclusions, the study highlights that scalability and sustainability are the main problems living labs encounter as open innovation intermediaries. | [85] | 95 |
| Open innovation and its effects on economic and sustainability innovation performance | The authors investigate the roles that different open innovation partners played in improving economic innovation performance and sustainability innovation performance. Authors found that economic innovation performance positively correlates with sustainability innovation performance, which implies that economic and sustainability innovation goals can be reached simultaneously. | [46] | 94 |
| Harnessing Difference: A Capability-Based Framework for Stakeholder Engagement in Environmental Innovation | Authors present a systematic review to enhance understanding of how firms can effectively incorporate stakeholder perspectives for environmental innovation. The study shows that engaging stakeholders in environmental innovation requires specific operational capabilities, engagement management capabilities, and capabilities to co-create innovative solutions and to learn from stakeholder engagement activities (systematized learning). | [88] | 80 |
| A systematic review of living lab literature | This study performs a systematic literature review of a sample of 114 scholarly articles about living labs to understand the central facets discussed in the nascent literature. Authors explore the origin of the living lab concept and its key paradigms and characteristics, including stakeholder roles, contexts, challenges, main outcomes, and sustainability. Living labs include heterogeneous stakeholders and apply various business models, methods, tools, and approaches. The benefits of living labs include tangible and intangible innovation and a broader diversity of innovation. | [103] | 68 |
| Consumer motives for peer-to-peer sharing | Authors develop a theoretical model to investigate the relative importance of consumer motives for and against peer-to-peer sharing. They validate the model through a survey finding that financial benefits, trust in other users, modern lifestyle, effort expectancy, and ecological sustainability are the most important drivers and prerequisites of platform usage intentions. | [104] | 64 |

**Table 10.** *Cont.*

| Document Title | Topics | Authors | Cites |
|---|---|---|---|
| Fostering sustainability by linking co-creation and relationship management concepts | This study analyzes the combination of co-creation and relationship management approaches with respect to sustainability through an exploratory multiple case study design. This study highlights the lack in integrating sustainability co-creation and relationship management and integrating different stakeholders to minimize negative social and ecological impacts. Authors suggest that sustainability relationship management must be anchored in a specific department and that the way of interacting with stakeholders have an influence on the process' outcome and the sustainability impact. | [105] | 64 |
| Simulation of B2C e-commerce distribution in Antwerp using cargo bikes and delivery points | They demonstrate how the effects of different e-commerce delivery concepts can be quantified with a simulation study using a real-world dataset and realistic cost values. Authors suggest that operational costs of companies can be reduced by stimulating customer self-pick-up, while externalities decrease with the implementation of a cargo bike distribution system. A sustainable solution and minimization of operational and external costs can be achieved involving stakeholders from industry and the public look. | [50] | 61 |
| Technology convergence, open innovation, and dynamic economy | This paper explores the link between open innovation, convergence and economic innovation that will come to the Fourth Industrial Revolution. Authors formulate policies for the technological, industrial, and economic orientation to alleviate the global economic crisis based on dynamic economy. | [34] | 60 |
| Holistic Innovation: An Emerging Innovation Paradigm | This paper systematically reviews the current typical innovation paradigms worldwide and their shortcomings, and introduces a new paradigm of innovation, holistic innovation, defined as total and collaborative innovation driven by strategic vision. Holistic innovation is a complex of strategic innovation, collaborative innovation, total innovation, and open innovation, which reflects wisdom from the Chinese context and Eastern culture. Holistic innovation helps China's enterprises build global innovation leadership and improves national innovation ability and optimizes the innovation policy design and action mindset to achieve global peace and sustainable development. | [106] | 59 |
| Radical Innovation for Sustainability: The Power of Strategy and Open Innovation | Authors present an in-depth case study of a sustainability-oriented innovation process for a radical new product within a multinational life sciences company, DSM. The study identifies five critical organizational practices through which strategic direction has enabled the innovation process: technology super-scouting throughout the value chain, search heuristics that favor radical sustainability solutions, integration of sustainability performance metrics in product development, championing the value chain to build demand for radical sustainability-oriented product innovation, and harnessing the benefits of open innovation. | [107] | 54 |
| Achieving sustainable e-commerce in environmental, social and economic dimensions by taking possible trade-offs | The purpose of this study is to integrate three dimensions (environmental, social, and economic) into e-commerce to ensure sustainability. This study collects empirical data from a case study involving companies in Kenya and Jordan. Authors suggest that all stakeholders in the virtual market must take appropriate responsibility since integration is essential for the sustainability of e-commerce in its three dimensions. Trade-offs must be taken in the various to realize sustainable e-commerce. | [15] | 52 |

**Table 10.** *Cont.*

| Document Title | Topics | Authors | Cites |
|---|---|---|---|
| Shared mobility for last-mile delivery: Design, operational prescriptions, and environmental impact | This paper provides new logistics planning models involving open-loop car routes, car drivers' wage-response behavior, interplay with the ride-share market, and optimal sizes of service zones within which passenger vehicles pick up goods and fulfill the last-mile delivery. Authors suggest that crowdsourcing shared mobility is not as scalable as the conventional truck-only system in terms of the operating cost. A transition to this paradigm has the potential for creating economic benefits by reducing the truck fleet size and exploiting additional operational flexibilities (e.g., avoiding high-demand areas and peak hours, adjusting vehicle loading capacities, etc.). | [108] | 52 |
| The diffusion of consumer innovation in sustainable energy technologies | This study investigates how consumer created technology solutions are diffused, and the role of prosumers (consumers participating in the product/service design process). Authors highlight that prosumers' efforts to diffuse their solutions remain low level and indicate directions for platform development by which prosumer solutions may spread more widely. | [109] | 51 |
| Sustainable retailing in the fashion industry: A systematic literature review | Authors identify the main perspectives of research on sustainable retailing in the fashion industry. As a result, the most prominent areas in the field are sustainable retailing in disposable fashion, fast fashion, slow fashion, green branding, and eco-labeling; retailing of secondhand fashion; reverse logistics in fashion retailing; and emerging retailing opportunities in e-commerce. | [110] | 50 |
| Relationship between convenience, perceived value, and repurchase intention in online shopping in Vietnam | This study examined the direct and indirect influence of the dimensions of online shopping convenience on repurchase intention through customer-perceived value in Vietnam. The results determined that the five dimensions of online shopping convenience are: access, search, evaluation, transaction, and possession/post-purchase convenience. All dimensions have a direct impact on perceived value and repurchase intention. | [111] | 50 |
| Can profit and sustainability goals co-exist? New business models for hybrid firms | This paper aims to discuss innovative business models for hybrid organizations aiming to embrace multiple and competing yet potentially synergistic goals (corporate sustainability and profit). Authors state that profit is the goal of traditional businesses' mission, but by making profit their only mission, firms risk missing out on the hidden opportunities latent in antagonistic assets. | [112] | 50 |

The main issues addressed by the most cited documents in the medium-term relate to sustainability achieved through open innovation [31,46,54,88,107]; sustainable e-commerce and retailing [15,110]; e-commerce delivery solutions [50,108]; online shopping and repurchase intentions [111]; open innovation and living labs [85,103]; open innovation, technologies and economic innovation [34]; the diffusion of consumer-created technology [109]; co-creation and sustainability [105]; holistic innovation and sustainable development [106]; big data and sustainable development [81]; peer-to-peer sharing in online platform operators [104]; sustainable business models [27]; corporate sustainability and profit [112]; and innovation in the agriculture and food industries [82].

Regarding the most recurrent research topics in the last five years, the results show that electronic commerce rises in position by being the second most discussed concept in the documents, and the open innovation topic falls to third place, as shown in Figure 6. Likewise, the results show that e-commerce changes position with innovation in the medium term compared to the long term, along with the concepts of big data, supply chain management, environmental impact, and business development. Economics, regional planning, competitive advantages, and supply chains represent the concepts lacking relevance in the medium term compared with the long term.

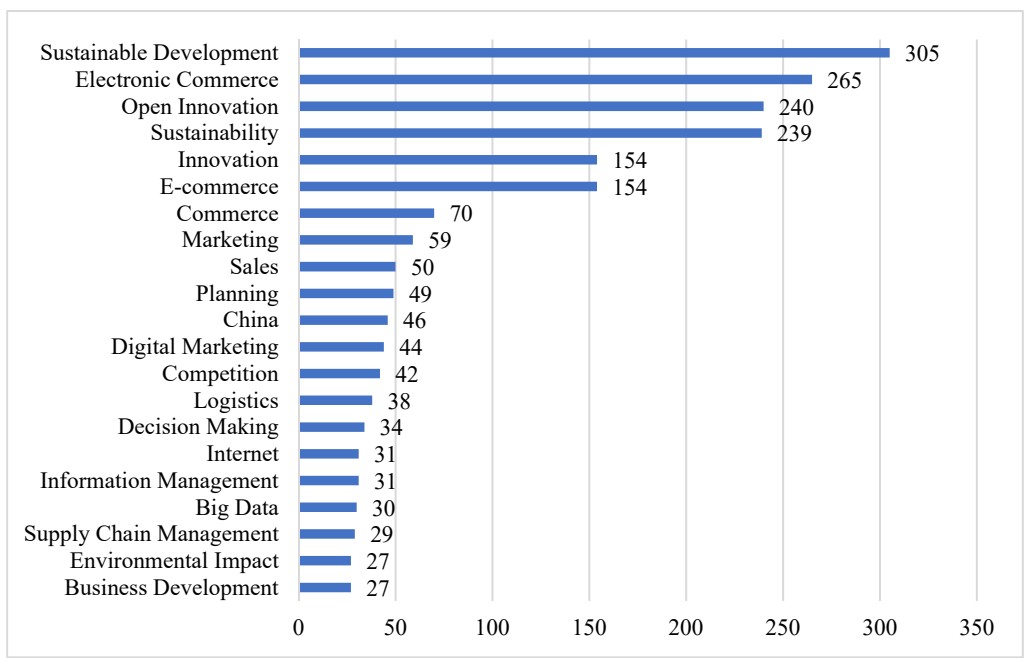

**Figure 6.** Main concepts in 2017–2021.

The association of these concepts is detailed in Figure 7, which identifies clusters around the main topics. In Cluster 1 (the red cluster), sustainability is identified as the central node, around which topics such as innovation, marketing, China, logistics, the Internet, and business development are clustered. In Cluster 2 (the green cluster), the main node is sustainable development, grouping terms such as open innovation, commerce, sales, planning, digital marketing, competition, and decision making. The central node of Cluster 3 (the blue cluster) is e-commerce, relating topics such as logistics and environmental impact. Electronic commerce is the central node of Cluster 4 (the yellow cluster), grouping concepts such as information management and big data. Cluster 5 (the purple cluster) focuses on ecosystems, co-creation, crowdsourcing and entrepreneurship. Cluster 6 (the light-blue cluster) focuses on consumption behavior and consumer behavior. Cluster 7 (the orange cluster) focuses on supply chain management, and is surrounded by concepts that are different from those in Figure 6, such as e-commerce business and technological change.

### 3.3. Short-Term Bibliometric Analysis: 2020–2021

The documents from the last two years (660 documents) represent 43.3% and 58.4% of the documents published in the last ten and five years, respectively. This implies that topics such as e-marketplaces, open innovation, and sustainability gained relevance in the scientific literature in recent years. As it happened in the medium term, social sciences continue to lead in the subject areas in the short term, while environmental science moves to second position after being in the third position, presenting continuous growth as a study area in the last ten years. Energy appears in the third position in Table 11, after being in the fourth position in the medium term and the sixth position in the long term, presenting continuous growth as a study area in the last ten years. Business, management and accounting fall to the fifth position after occupying the second position in the medium term and the first position in the long term. This implies that the documents published in recent years are more related to the social sciences, the environment, and energy use.

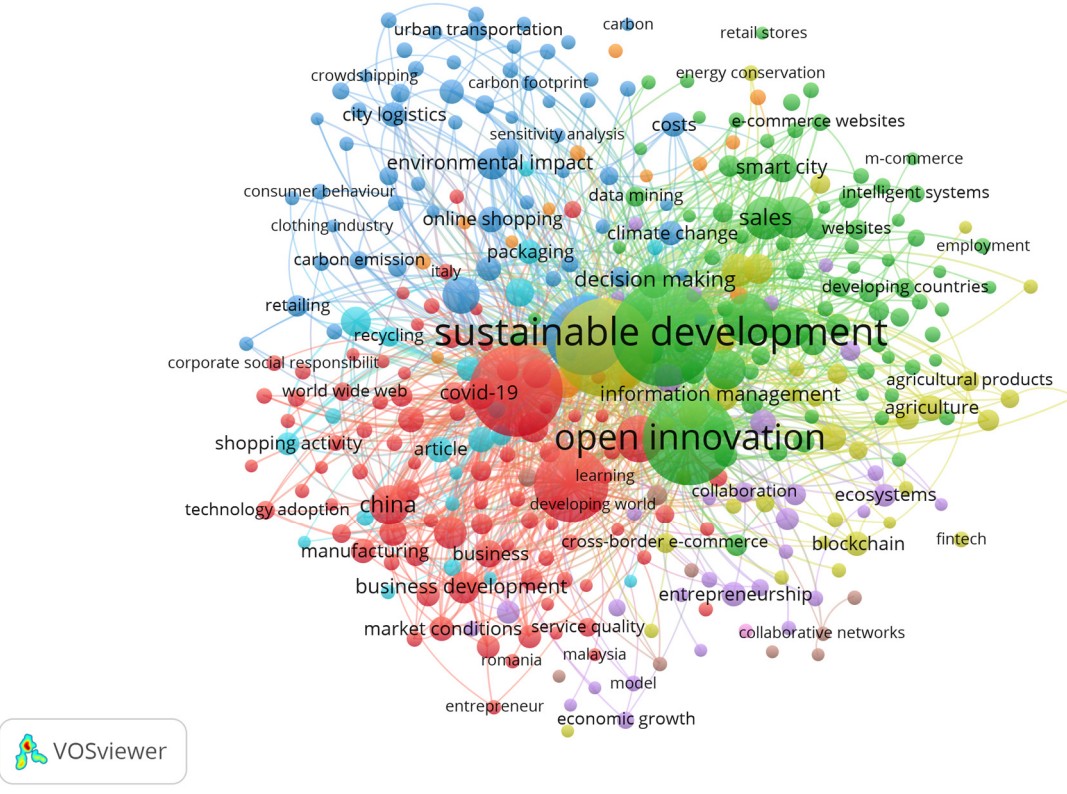

**Figure 7.** Co-occurrence analysis in 2017–2021.

**Table 11.** Subject areas for 2020–2021.

| Subject Area | Document | % Ind. | % Accum. |
|---|---|---|---|
| Social Sciences | 275 | 18.0% | 18.0% |
| Environmental Science | 209 | 13.7% | 31.7% |
| Energy | 181 | 11.8% | 43.5% |
| Computer Science | 176 | 11.5% | 55.0% |
| Business, Management and Accounting | 173 | 11.3% | 66.4% |
| Engineering | 134 | 8.8% | 75.1% |
| Economics, Econometrics and Finance | 103 | 6.7% | 81.9% |
| Decision Sciences | 83 | 5.4% | 87.3% |
| Others | 161 | 12.6% | 100% |

For the leading journals for 2020–2021, which have accumulated 39.2% of the publications of the last two years, Table 12 shows that Sustainability Switzerland and Journal of Open Innovation Technology Market and Complexity occupy the first two positions, as happened for the medium and long terms, indicating that the greatest volume of publications about this research is through journals attached to the publisher MDPI (Multidisciplinary Digital Publishing Institute). E3s Web of Conferences rises to the third position after being in the fifth position in the medium term, while Journal of Cleaner Production moves from the third position to the fifth position in the last two years. Advances In Intelligent Systems and Computing remains in the fourth position. For the short term, the International Journal of Logistics Research and Applications, Lecture Notes in Networks and Systems, and Sustainable Cities and Society appear as the leading journals, with contributions of five documents each. IFIP Advances in Information and Communication Technology and

IOP Conference Series Materials Science and Engineering disappear from the ranking of Table 12.

**Table 12.** Leading journals for 2020–2021.

| Journal | Docs. | % Docs. | Publication Type | *h*-Index 2020 (Scimago) | Max Quartil 2020 (Scimago) |
|---|---|---|---|---|---|
| Sustainability Switzerland | 110 | 16.7% | Journals | 85 | Q1 |
| Journal of Open Innovation Technology Market and Complexity | 47 | 7.1% | Journals | 22 | Q2 |
| E3s Web of Conferences | 18 | 2.7% | Conferences and Proceedings | 22 | N/A |
| Advances in Intelligent Systems and Computing | 16 | 2.4% | Book Series | 41 | N/A |
| Journal of Cleaner Production | 14 | 2.1% | Journals | 200 | Q1 |
| IOP Conference Series Earth and Environmental Science | 11 | 1.7% | Conferences and Proceedings | 26 | N/A |
| Lecture Notes in Computer Science | 9 | 1.4% | Book Series | 400 | Q3 |
| Smart Innovation Systems and Technologies | 8 | 1.2% | Book Series | 22 | Q4 |
| Journal of Physics Conference Series | 6 | 0.9% | Conferences and Proceedings | 85 | Q4 |
| ACM International Conference Proceeding Series | 5 | 0.8% | Conferences and Proceedings | 123 | N/A |
| International Journal of Logistics Research and Applications | 5 | 0.8% | Journals | 33 | Q1 |
| Lecture Notes in Networks and Systems | 5 | 0.8% | Book Series | 14 | Q4 |
| Sustainable Cities and Society | 5 | 0.8% | Journals | 61 | Q1 |

The leading authors shown in Table 13 indicate that Costa presents the highest number of publications for the medium and short terms, generating articles related to e-commerce [113], open innovation, and sustainability [36]. Other authors who also remain in the ranking of scientific production in the short and medium terms are Buldeo Rai, Dang, Wang, Abreu, Gatta, Macharis, Marcucci, and Zhao. Authors such as Nguyen, Medina-Salgado, Settembre-Blundo, and Ekren appeared in the last two years, contributing to the publication of three articles each. Nguyen publishes in co-authorship with Dang and Wang on logistics and e-commerce logistics [95,114]. Medina-Salgado and Settembre-Blundo co-author on cybersecurity and sustainability in e-commerce [115,116], and open innovation and life cycles [117]. Ekren researches sustainable e-commerce networks [118,119]. Moreover, Costa, Gatta, and Marcucci remain in the ranking of the most relevant authors of the last 10, 5 and 2 years.

**Table 13.** Leading authors for 2020–2021.

| Author | Docs. * | Scopus Author ID | *h*-Index and Citations | Main Subject Area | Affiliation | Country |
|---|---|---|---|---|---|---|
| Costa, J. | 6 | 57212821686 | *h*-index: 5 74 citations by 68 documents | Open Innovation; Alliance Portfolios; Absorptive Capacity; Community Innovation Survey; Marketing Innovation; Manufacturing Firms | Universidade de Aveiro | Aveiro, Portugal |
| Buldeo Rai, H. | 4 | 57195135422 | *h*-index: 12 260 citations by 217 documents | Urban Freight Transport; City Logistics; Cargo | Vrije Universiteit Brussel | Brussels, Belgium |
| Dang, T.T. | 4 | 57218565464 | *h*-index: 6 113 citations by 73 documents | Decision-making, e-commerce marketplaces, Logistics, Last Mile Delivery | International University, Vietnam National University Ho Chi Minh City | Ho Chi Minh City, Viet Nam |
| Wang, C.N. | 4 | 7501640993 | *h*-index: 17 1170 citations by 884 documents | Decision-making, e-commerce marketplaces, Logistics, Last Mile Delivery | National Kaohsiung University of Science and Technology | Kaohsiung, Taiwan |
| Abreu, A. | 3 | 57218315486 | *h*-index: 12 380 citations by 252 documents | Zero Energy Buildings; Refurbishment; Renovation; Alliance Portfolios; Absorptive Capacity; Open Innovation | Instituto Superior de Engenharia de Lisboa | Lisbon, Portugal |
| Ekren, B.Y. | 3 | 23488489800 | *h*-index: 18 1355 citations by 858 documents | Green Supply Chain Management; Environmentally Preferable Purchasing; Green Practices; Lateral Transshipment; Spare Parts; Inventory Systems | Yaşar Universitesi | Izmir, Turkey |
| Gatta, V. | 3 | 35109007100 | *h*-index: 24 1191 citations by 558 documents | Urban Freight Transport; City Logistics; Cargo | Università degli Studi Roma Tre | Rome, Italy |
| Macharis, C. | 3 | 6507193118 | *h*-index: 39 5524 citations by 3944 documents | Urban Freight Transport; City Logistics; Cargo | Vrije Universiteit Brussel | Brussels, Belgium |
| Marcucci, E. | 3 | 6602255083 | *h*-index: 28 1912 citations by 1068 documents | Urban Freight Transport; City Logistics; Cargo | Høgskolen i Molde | Molde, Norway |
| Medina-Salgado, M.S. | 3 | 57216226774 | *h*-index: 4 70 citations by 63 documents | Business Model Innovation; Sustainable Business; Digital Transformation | Universidad Rey Juan Carlos | Madrid, Spain |
| Nguyen, N.A.T. | 3 | 57218570813 | *h*-index: 6 75 citations by 56 documents | Decision-making, e-commerce marketplaces, Logistics, Last Mile Delivery | National Kaohsiung University of Science and Technology | Kaohsiung, Taiwan |
| Settembre-Blundo, D. | 3 | 57194205987 | *h*-index: 12 418 citations by 328 documents | Social Life Cycle Assessment; United Nations Environment Program; Social Indicators | Universidad Rey Juan Carlos | Madrid, Spain |
| Zhao, X. | 3 | 57193208482 | *h*-index: 14 567 citations by 419 documents | Open Innovation; Absorptive Capacity; Business Model Innovation; Sustainable Business; Digital Transformation | Daegu Gyeongbuk Institute of Science and Technology | Daegu, South Korea |

* Documents published in Scopus in 2020–2021 related to open innovation, sustainability, and e-marketplaces.

Figure 8 shows the countries producing more than ten documents in the last two years, highlighting that China maintains the leadership in publications for the long, medium, and short terms, contributing 14.6% of the documents published in the last two years (131 documents). India moves up one position to rank second, and Italy and Spain rank fourth, contributing 44 documents each (4.9%). The United States falls from second place in the medium term to sixth place in the short term, and the United Kingdom retains fifth place. Vietnam appears in the ranking with 13 documents, while Romania, Finland, and France lose relevance by contributing ten or fewer publications in this period. The leading countries in Asia, in Figure 8, contribute 30.4% of the total scientific production in e-marketplaces, open innovation, and sustainability, and the leading European countries contribute 29.1%. Unlike the analysis of the long and medium terms, the leading countries of Asia surpass, in terms of scientific production, those from Europe due to the high development of e-markets and e-commerce in Asian countries, especially in China.

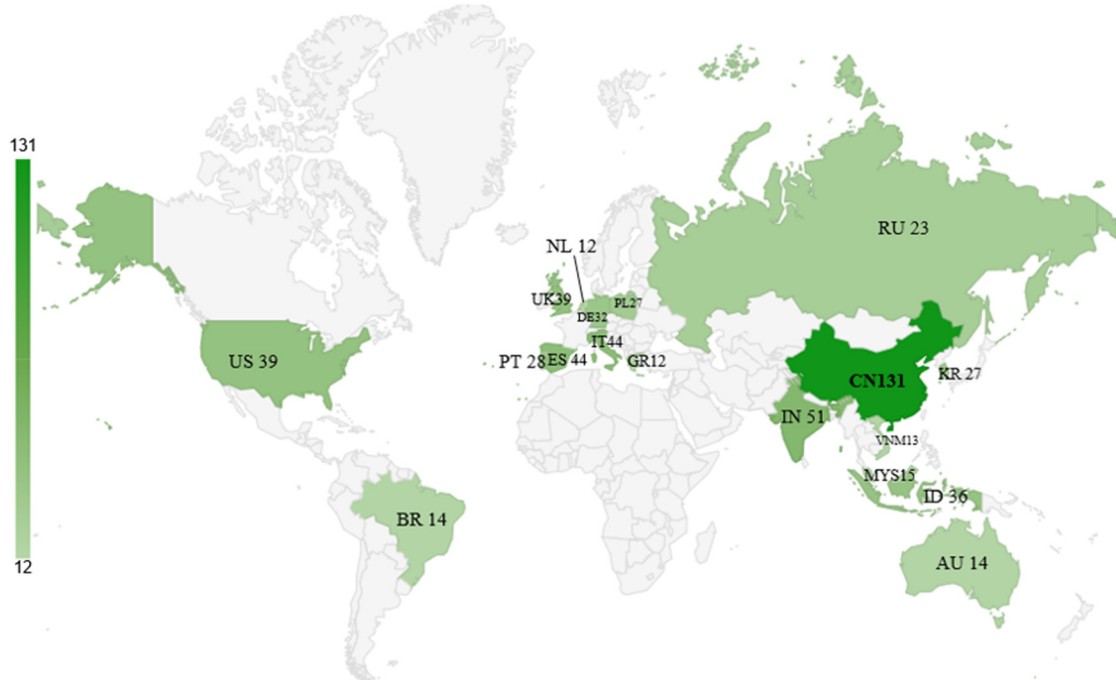

**Figure 8.** Scientific output by country/territory in 2020–2021.

As shown in Table 14, the Universidade de Aveiro and Universidad Rey Juan Carlos maintain the first and second place, respectively, in the short term as leading affiliations, largely thanks to the contributions of Costa (Universidade de Aveiro) and the publications of Medina-Salgado and Settembre-Blundo (Universidad Rey Juan Carlos). Universidad de Málaga appears in the third place, while the Politecnico di Milano disappears from the leading affiliations. Other institutions that appear in the ranking are the Institute for Systems and Computer Engineering, Technology and Science; Lusófona University; the Financial University under the Government of the Russian Federation; and Universidade de Lisboa. On the other hand, the following disappear from the leading affiliations: Tecnologico de Monterrey, Politecnico di Milano, Queensland University of Technology, Technical University of Berlin, Daegu Gyeongbuk Institute of Science and Technology, and Bucharest University of Economic Studies. Università degli Studi di Napoli Federico II and Università degli Studi Roma Tre largely explain Italy's rise in the ranking by country. Likewise, the Rey Juan Carlos University, the University of Malaga, and the Polytechnic University of Madrid largely explain the rise of Spain in the ranking by country. Portugal ranks ninth in the ranking of countries supported by affiliations such as Universidade de

Aveiro; the Institute for Systems and Computer Engineering, Technology and Science; and Lusófona University.

**Table 14.** Leading affiliations for 2020–2021.

| Affiliation | Documents | % Ind. | % Accum. |
|---|---|---|---|
| Universidade de Aveiro | 11 | 2.5% | 2.5% |
| Universidad Rey Juan Carlos | 9 | 2.1% | 4.6% |
| Universidad de Málaga | 6 | 1.4% | 6.0% |
| Peter the Great St. Petersburg Polytechnic University | 6 | 1.4% | 7.3% |
| Università degli Studi di Napoli Federico II | 6 | 1.4% | 8.7% |
| Institute for Systems and Computer Engineering, Technology and Science | 6 | 1.4% | 10.1% |
| Bina Nusantara University | 6 | 1.4% | 11.5% |
| Lusófona University | 5 | 1.1% | 12.6% |
| Università degli Studi Roma Tre | 5 | 1.1% | 13.8% |
| Universidad Politécnica de Madrid | 5 | 1.1% | 14.9% |
| Financial University under the Government of the Russian Federation | 5 | 1.1% | 16.1% |

The three most cited documents in the short term received an average of 47 citations per year [9], 20 citations per year [17], and 19 citations per year [120]. Table 15 shows that 25% of the most cited documents belong to Sustainability (Switzerland), with an average of 23.4 citations in two years and 11.7 citations per year; on the other hand, 15% of the most cited documents belong to the Journal of Cleaner Production, with an average of 29.3 citations in two years and 14.6 citations per year. Finally, 10% of the most cited documents belong to the Journal of Open Innovation: Technology, Market, and Complexity, with an average of 17.5 citations in two years and 8.7 citations per year.

**Table 15.** Most cited documents in 2020–2021.

| Document Title | Topics | Authors | Cites |
|---|---|---|---|
| Managing the effectiveness of e-commerce platforms in a pandemic | This study presents a systematic framework to examine the effect of the perceived effectiveness of e-commerce platforms (PEEP) on consumer's perceived economic benefits in predicting sustainable consumption. This study finds a positive moderating effect of pandemic fear on the relationships among PEEP, economic benefits, and sustainable consumption. | [9] | 47 |
| A multiobjective optimization model for sustainable reverse logistics in Indian E-commerce market | This paper proposes a multi-objective logistics network model for the return products of the Indian e-commerce market. Authors propose a multi-objective optimization on the three fronts of sustainability: economical (cost), environmental (environmental impact of different process), and social (workdays created and lost due to harms at work). This study will help the managers in deciding the number of facility stores, warehouses and technologies needed to operate. | [17] | 40 |
| Acceptance of autonomous delivery vehicles for last-mile delivery in Germany—Extending UTAUT2 with risk perceptions | This study investigates the users' acceptance of Autonomous delivery vehicles (ADVs) in last-mile delivery in Germany. The results indicate that price sensitivity is the strongest predictor of user acceptance, followed by performance expectancy, hedonic motivation, perceived risk, social influence and facilitating conditions. | [120] | 39 |

**Table 15.** *Cont.*

| Document Title | Topics | Authors | Cites |
|---|---|---|---|
| Sustainability condition of open innovation: Dynamic growth of alibaba from SME to large enterprise | This article analyzes how Alibaba became a global top e-commerce company in China in a short time. Alibaba has applied global, creative e-commerce business models through open innovation in a short time. It has overcome the cost of open innovation and the force that breaks down a company through an open innovation-friendly culture and open business model feedback loop. | [35] | 35 |
| Analysis of the relationship between open innovation, knowledge management capability and dual innovation | Authors propose to an internal linkage framework for open innovation, knowledge management capability and dual innovation to ensure innovation and sustainability. Research results show that both open innovation and knowledge management capability have a positive influence on dual innovation (exploitation innovation and exploration innovation). Authors conclude that open innovation (inward- oriented and outward-oriented open innovation) influences dual innovation. | [38] | 32 |
| Business model, open innovation, and sustainability in car sharing industry-Comparing three economies | This paper discusses dynamics and differences of business models in the car-sharing industry and focuses on open innovation as the trigger of diverse business models among Uber in the U.S., DiDi Chuxing in China, and KakaoT in Korea. Authors study the differences in the business models of the car-sharing industry, identifying that open innovation strategies determine the contents and dynamics of car-sharing business models, such as the revenue business model, responsibility business model. | [28] | 31 |
| Analyzing barriers for adopting sustainable online consumption: A rough hierarchical DEMATEL method | This research aims at developing a method to identify and visualize the vague interrelationships among barriers for adopting sustainable online consumption. These barriers include backward sustainable production technology, lack of proactive plans to adopt sustainable production and consumption, similar types of products in offline store, lack of information about a product when shopping online, lack of policy support, lack of government regulations, lack of awareness of sustainable consumption, insufficient knowledge about sustainable consumption, low level of costumers' consumption. An application in a large e-commerce company shows the efficiency and effectiveness of the integrated method. | [22] | 31 |
| Evaluating the environmental impacts of online shopping: A behavioral and transportation approach | The authors develop an econometric behavioral model to understand the factors that affect shopping decisions, both in-store and online. The study estimates potential vehicle miles traveled and environmental emissions in two metropolitan areas, Dallas and San Francisco (SF) and estimates the impacts of rush deliveries, basket size, and consolidation levels. | [48] | 29 |
| Business model innovation through a rectangular compass: From the perspective of open innovation with mechanism design | Authors apply the open innovation concept to the design of creative business models. They built a rectangular compass concept model based on four aspects: over-shooting of modern business models, expanding the bottom of modern business models, cultivating the forward neighborhood of modern business models, and cultivating the backward neighborhood of modern business model. The study highlights that open innovation is the engine of sustainable business model innovation dynamics. | [121] | 25 |
| Closed-loop supply chain network design and modelling under risks and demand uncertainty: an integrated robust optimization approach | This study proposes a generic closed-loop supply chain (sum of sustainable activities like green purchasing, green manufacturing and material management, green distribution and marketing and also reverse logistics) network based on mixed integer programming formulation considering a total of four levels of uncertainty for four different networks. This approach helps supply chain managers to refine risk management strategies to handle risk events. | [122] | 25 |

**Table 15.** *Cont.*

| Document Title | Topics | Authors | Cites |
|---|---|---|---|
| Sustainable open innovation to address a grand challenge: Lessons from Carlsberg and the Green Fiber Bottle | This paper describes the case of how the Danish beer manufacturer, Carlsberg, developed the Green Fiber Bottle as part of its sustainability program through an open innovation approach in collaboration with complementary partners. It thereby illustrates how a grand challenge associated with sustainability can be effectively addressed through open innovation and reveals the opportunities and challenges that emerge in that context. | [123] | 25 |
| Open innovation 4.0 as an enhancer of sustainable innovation ecosystems | This study presents how open innovation can enhance sustainable innovation ecosystems and boost the digital transition. Authors trace a diachronic perspective of the sustainable innovation ecosystem, its connection to open innovation, and identification of the university linkages. They propose a policy package towards green governance, empowering the university in governance distributed ecosystem, embedded in the community, self-sustained with shared gains, and a meaningful sense of identity. | [36] | 25 |
| Sustainable B2B E-commerce and blockchain-based supply chain finance | This identifies trends in supply chain financing in China's B2B e-commerce and analyzes the introduction of blockchain technology in supply chain financing of Alibaba's B2B commerce platform. The main advantage of using blockchain is that it creates a decentralized database that is secure, it increases the speed of payment and the reliability and transparency of data transfer which can support the development of much more sustainable economies. | [51] | 23 |
| The two-echelon vehicle routing problem with covering options: City logistics with cargo bikes and parcel lockers | Authors propose sustainable applications for e-commerce and city distribution based on the two-echelon vehicle routing problem with covering options. This problem involves a single depot, parcel lockers, satellite locations, trucks and zero- emission vehicles (such as cargo bikes). This study suggests that the use of parcel lockers has a great potential to reduce driving distance. | [124] | 21 |
| Decentralized accessibility of e-commerce products through blockchain technology | This study proposes a blockchain-based solution that integrates the product chain and supply chain to provide a transparent and decentralized resource of product and its access information. This distributed and transparent approach for reducing the complexity for tracing the e-commerce products ensures the social and financial sustainability. | [125] | 20 |
| At the Epicenter of COVID-19–the Tragic Failure of the Global Supply Chain for Medical Supplies | Authors study the governance and resilience of the global supply chains for medical supplies en el context of COVID-19 pandemic. They propose a model and recommend a new governance system that supports intervention by public-health authorities during critical emergencies through new technologies such as advanced analytics and blockchain. These results minimize the compromise of our healthcare workers and health systems due to infection exposure and build capacity toward preparedness and action for a future outbreak. | [126] | 20 |
| Smart digital marketing capabilities for sustainable property development: A case of Malaysia | This study aims at understanding the principles and practices of sustainable digital marketing in the Malaysian property development industry. Authors propose a marketing technology acceptance model to investigate the adoption of digital marketing, the impediments to its adoption, and the strategies to improve digital capabilities for the local context. The results show that the sample property development companies are driven by the benefit of easily obtaining real-time customer information for creating and communicating value to customers more effectively through the company brand. | [127] | 18 |

**Table 15.** *Cont.*

| Document Title | Topics | Authors | Cites |
|---|---|---|---|
| Sustainability in e-commerce packaging: A review | This paper reviews the evolution of packaging over the last century through a compilation of scientific literature on e-commerce packaging focusing on its environmental side. Authors highlight that some packaging products continue to be made from non-renewable materials and thus restrict growth of e-commerce and recommend further research in producing new packages from renewable sources such as cellulose-containing materials, or from recycled cellulose- based materials such as carton board to reduce the environmental impact of packaging. | [128] | 18 |
| Analyzing the critical success factor of CSR for the Chinese textile industry | This work studies the critical success factors of corporate social responsibility (a sustainable strategy) in textile industries in China. Government initiatives is the most influential common success factor of corporate social responsibility implementation in Chinese textile industries, followed by customer pressure, environment management system, and the improvement of human rights including law, safety, and wellness. | [129] | 17 |
| Economic growth, increasing productivity of SMEs, and open innovation | This study analyzes how economic growth works as a determinant of increasing the productivity of small and medium enterprises; the influence of government policies, business capital support, and the strengthening of human resource capacity on the development of small and medium enterprises (SMEs); and strategies to increase business productivity and the sustainability of SMEs. Authors recommend an economic growth strategy based on technological innovation to increase the productivity of community economic enterprises in Makassar City, Indonesia | [130] | 17 |
| Resilience effects in food consumption behaviour at the time of COVID-19: perspectives from Italy | This paper gives an overview of the recent changes in consumption patterns that occurred due to the Italian lockdown, and the evolution of the main food supply chains. Home delivery has been the most important element in this context, as it boomed during this period, helping laggard consumers fill the digital divide, as it was mostly mediated by e-commerce platforms and instant messaging. E-commerce platforms also leveraged small retailers and small producers to regain their space. | [131] | 17 |

The main issues addressed by the most cited documents are related to sustainability through open innovation [36,123]; business models and open innovation [28,121]; e-commerce logistics and sustainability [17,48,120,124,131]; the sustainability of global supply chains [126]; supply chain design and risks [122]; e-commerce, open innovation and sustainability [35]; the effectiveness of e-commerce platforms [9]; barriers for adopting sustainable online consumption [22]; blockchain technology and e-commerce [51,125]; sustainability in e-commerce packaging [128]; open innovation and dual innovation [38]; success factors of corporate social responsibility [129]; digital marketing for sustainable property development [127]; and open innovation, economic growth and SMEs [130].

Figure 9 describes the main concepts addressed in the publications of the last two years, finding that sustainable development and electronic commerce are the most recurrent topics. The sustainability concept increases its relevance concerning open innovation. E-commerce, being directly related to the concept of electronic commerce, retains fifth place. Regarding the main concepts of the medium term, information management, the Internet, environmental impact, and business development disappear in the short term, and the concepts of COVID-19, consumption behavior, and supply chains appear, focusing on the problems generated by the pandemic, the supply of products, and changes in consumer behavior towards electronic platforms.

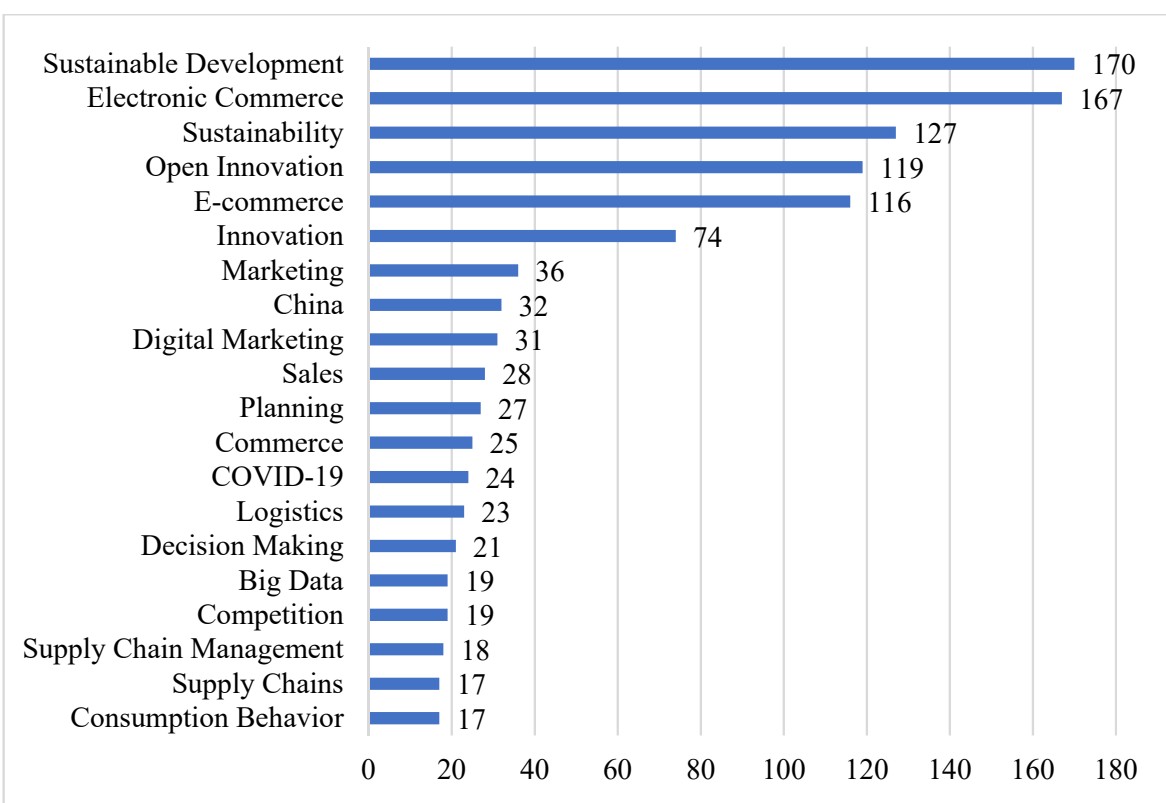

**Figure 9.** Main concepts in 2020–2021.

The association of these concepts is detailed in Figure 10, which identifies clusters around the main topics. In Cluster 1 (the red cluster), sustainability is identified as the central node, around which topics such as innovation, China, logistics, and supply chain management are clustered. Cluster 2 (the green cluster) presents the term "e-commerce" as the central node, surrounded by concepts such as sales, planning, decision-making, competition, and consumption behavior. The central node of Cluster 3 (the blue cluster) is electronic commerce, relating to topics of marketing, big data, and supply chains. E-commerce is the central node of Cluster 4 (the yellow cluster), grouping terms—like consumption behavior—which are different from those in Figure 9, such as environmental impact, city logistics, and environmental sustainability. Cluster 5 (the purple cluster) focuses on COVID-19, surrounded by topics which are different from those in Figure 9, such as the Internet, commercial phenomena, and shopping activity. Cluster 6 (the light-blue cluster) focuses on open innovation, grouping terms which are different from those in Figure 9, such as digital platforms, co-creation, collaboration, and crowdsourcing. In Cluster 7 (the orange cluster), the main node is sustainable development, grouping terms like digital marketing and commerce, and other concepts such as business sustainability, business models, and digital transformation.

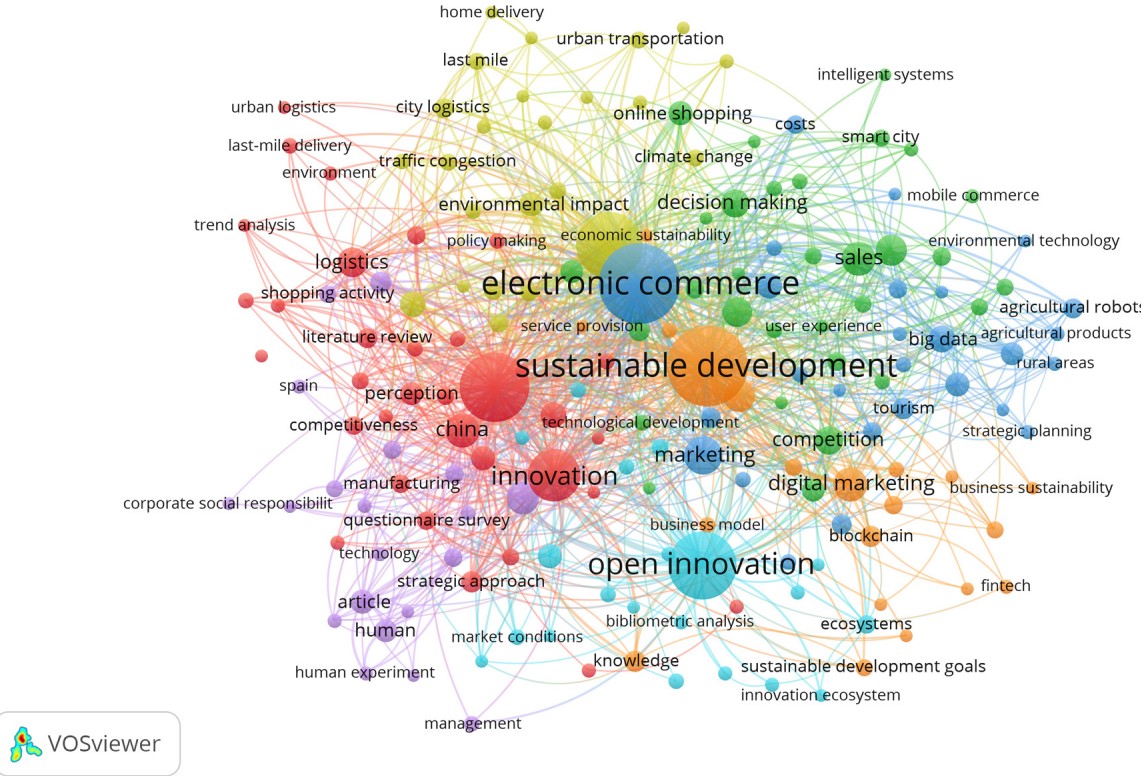

**Figure 10.** Co-occurrence analysis in 2020–2021.

## 4. Discussion

The analysis presented in the previous sections shows that publications related to e-marketplaces, open innovation, and sustainability have grown exponentially in the last ten years. It highlights the participation of publications in the short term (the last two years), which allows us to infer that this theme is of great relevance in current times for the scientific and academic community, enabling multiple lines of research to be extended around each aspect of said theme. Research lines in subject areas such as social sciences, environmental sciences, energy, business, management, and accounting have been of great interest in recent years. Specifically, the social science area investigates in-depth topics related to open innovation, and the environmental sciences and energy areas address the issue of sustainability to a large extent. The dynamics of business models of e-marketplaces are usually addressed through the business, management, and accounting areas. The topics addressed by the authors with the most contributions are related to sustainability and open innovation, and their impact on business models; education and knowledge innovation towards sustainable development; e-commerce, open innovation, and sustainability; sustainable supply chains and e-commerce logistics; sustainability and environmental impact; and open innovation in SMEs. The most recognized authors in the periods analyzed are Park, Ramírez-Montoya, Yun, Callou, Saguy, Costa, Gatta, Marcucci, Buldeo Rai, Dang, and Wang.

Among the journals that contributed the most documents on the research topic, Sustainability Switzerland stands out, having gone from publishing an average of 20 articles per year in the last ten years to publishing 55 documents per year in the past two years due to its approach to issues related to sustainability from a multidisciplinary point of view. Likewise, the Journal of Open Innovation Technology Market and Complexity stands out, having published an average of 23.5 documents related to open innovation per year in the last two years. The Journal of Cleaner Production occupied between the third and fifth place in terms of the journals with the greatest contribution of documents in the findings of this investigation; however, this journal presents more citations per year in its articles

than the two journals mentioned above, for which the articles from the Journal of Cleaner Production have a high impact according to Scopus. Sustainability Switzerland follows the Journal of Cleaner Production in the average number of citations per year for the most relevant papers. Other publications such as E3s Web of Conferences and Advances in Intelligent Systems and Computing provide sufficient documents primarily derived from proceedings of important conferences, symposia, and congresses.

China ranks first in the periods analyzed as the country that generates the greatest number of documents on the research topic of this study, publishing more than twice as many documents as the United States and India in the medium and short terms, respectively. India, Italy, and Spain presented remarkable changes for the last two years, providing more documents than the United Kingdom and the United States. Likewise, the findings indicate that the accumulated production of documents has been co-led mainly by Asian and European countries. The rise of e-commerce platforms, the development of e-marketplaces, the growth of companies and business models, and the global pressures to guarantee sustainable development have made China the largest global producer of goods, as it is interested in researching and publishing on topics related to e-marketplaces, open innovation, and sustainability.

The positioning of the leading countries is also explained through some leading affiliations. In the case of Spain in the medium and short terms, the Rey Juan Carlos University and the University of Malaga stand out. For Italy, the Politecnico di Milano (to which leading authors such as Mangiaracina and Tumino belong), Università Degli Studi Roma Tre (to which leading authors such as Gatta belong), and Università Degli Studi di Napoli Federico II represent important affiliations. Likewise, Universidade de Aveiro (to which leading authors such as Costa belong) and the Institute for Systems and Computer Engineering, Technology and Science stand out in Portugal. Other important affiliations include Daegu Gyeongbuk Institute of Science and Technology (South Korea, home to leading authors such as Yun and Zhao), Bina Nusantara University (India), Peter the Great St. Petersburg Polytechnic University (Russia), Bucharest University of Economic Studies (Romania), and Delft University of Technology (Netherlands).

The documents with the highest impact address topics such as e-commerce and environmental sustainability, sustainability achieved through open innovation, sustainable business models, business models and open innovation, sustainable e-commerce and retailing, e-commerce logistics and sustainability, supply chain sustainability, innovation and living labs, open innovation and smart cities. This indicates that the main emphasis is placed on environmental impacts by combining concepts of e-commerce/e-marketplaces with sustainability, specifically through logistics operations related to transportation, distribution, and delivery. As for open innovation, this tends to improve business models and social systems in living labs and cities.

Figure 11 presents the evolution of the 20 most relevant concepts and themes over the last ten years. In this sense, Sustainable Development represents the main concept around e-marketplaces, open innovation, and sustainability, as it integrates multiple dimensions in order to guarantee the needs of the present without compromising future generations, implying a relationship between economic growth and the environment. Likewise, the sustainability concept—which contributes to balancing the social, economic, and environmental pillars—has become more relevant in the last two years. Regarding the difference between sustainable development and sustainability, sustainable development focuses on the development concept [132], whereas sustainability focuses on the environment concept [133]. Authors such as Shaker [134] suggest that sustainability represents the goal, while sustainable development describes the process to achieve this goal. In a business context, corporate sustainability is measured by different international standards such as the Global Reporting Initiative (GRI) and Dow Jones Sustainability (DJS), which evaluate different dimensions, such as environmental, economic, and social dimensions [135].

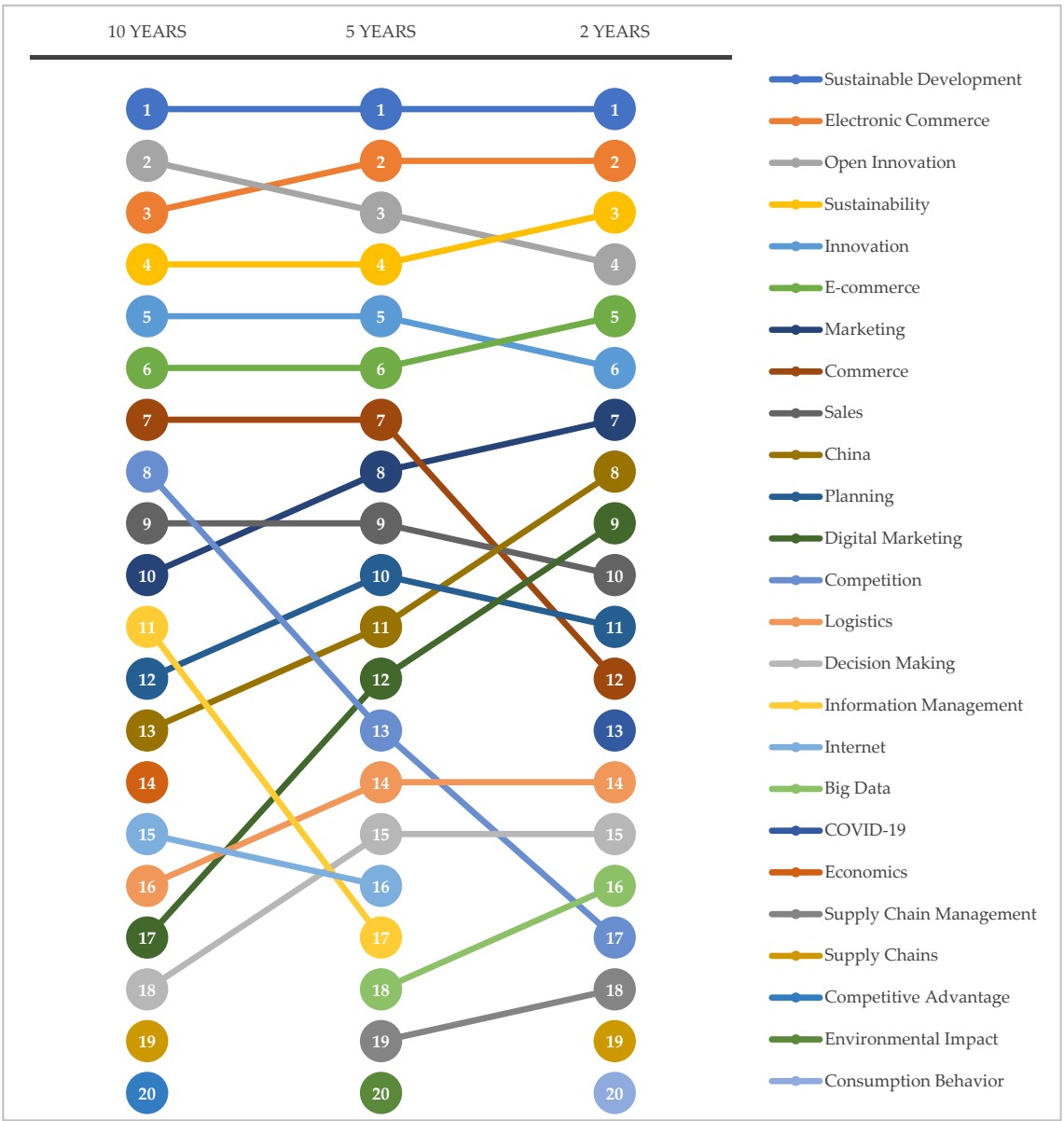

**Figure 11.** Evolution of the concepts by period.

The concepts of electronic commerce and e-commerce refer to the process of buying and selling products or services electronically [113], where E-commerce represents the abbreviation of electronic commerce. These concepts have increased their relevance, especially in the last five and two years, due to the rise of digital platforms that improve conditions for buyers and sellers, reduce environmental impacts related to mobility in cities, and react to the effects of lockdowns caused by the COVID-19 pandemic. Therefore, the concepts of Open Innovation and Innovation lose their relevance in the short term; however, they continue to be of great importance for the integration and collaboration of stakeholders for business model improvement, especially those based on digital platforms. Regarding the difference between innovation and open innovation, the traditional approach to innovation has been supported by the generation and development of ideas arising from and treated within R&D departments [136], which is insufficient because it leads to dependence on the creation of successful ideas and the hiring of experts in the business units that require innovation activities [29], which in turn results in more expense for a company. Open innovation requires both inside-out and outside-in interactions from the company, which are more efficient than one-way interaction [137]. Open innovation

implies a dynamic company interaction with external stakeholders such as customers, users, suppliers, universities, research and development centers, competitors, government agencies, the surrounding communities, and society [30].

The increasing relevance of concepts such as marketing, digital marketing, China, and logistics is striking, and the emergence and increase in the last five years of big data and supply chain management stand out. Marketing and digital marketing are gaining relevance due to the impact of digital media on the performance and revenue growth of SMEs, as well as the emergence of business opportunities and business models based on electronic platforms. For its part, China represents the country with the greatest number of authors' institutional affiliations; as such, it is to be expected that the focus of many investigations will be related to or directed to that country. Logistics is directly related to the environmental impact of e-commerce and its contribution to the sustainability of e-marketplace platforms in the purchasing, transport, and delivery process. The big data concept has focused on environmental pollution and prevention using large-scale data in order to enable profitability and sustainability through strategic operations and marketing-related business activities. The relationship between big data, digital devices, and infrastructure allows the management of large amounts of data in cities. Supply Chain Management is increasing in relevance due to the focus on information transparency and appropriate communication between partners in order to guarantee sustainability with technologies such as blockchain, and to improve logistics operations.

Likewise, the importance of the COVID-19 concept in the last two years is related to mobility restrictions and lockdowns because of the COVID-19 pandemic, which generated an online migration in which consumers are increasingly turning to online purchases through e-commerce platforms. Some concepts that maintain their relevance in the periods analyzed are sales, planning, and decision making. The sales concept represents the main objective promoted by e-marketplaces, as well as the main means to achieve economic sustainability of e-marketplaces. Planning and decision making are related to each other regarding the facilitation of policy-making definition and policy deployment to find ways to make e-commerce sustainable. Moreover, the concepts of competition, commerce, and information management have lost some relevance over the last ten years, but continue to be in the top 20.

Based on the results of this study, the consistent growth of e-commerce in the last decade has altered customers' shopping experiences, causing more trucks than ever before to enter cities, bringing with them the negative externalities of increased congestion and pollution [48]. This behavior is expected to continue to grow, because according to experts, by 2026, nearly 40% of all products globally will be sold online [138]. Therefore, a cleaner and sustainable environment is becoming the topmost priority for both owners and stakeholders involved in e-businesses [17].

Because the environmental, social, and economic aspects are significant to the e-commerce sector on both the retailer and consumer sides, they must be treated jointly to create economic value and achieve benefits in policy-making and environmental protection [15]. We can mention some alternatives for the achievement of sustainability in e-marketplaces, such as the improvement of consumer confidence in electronic platforms [15], attracting and retaining sellers to secure the platform's long-term viability and success [139], the development of new delivery practices in last-mile logistics in order to foster a sustainable urban environment based on stakeholder objectives [48,120,140], the increase in profit by exploiting the residual capacity of vehicles [141], the reduction of externalities like traffic congestion or emissions without implying higher costs for companies [50], and the development of information and communication for the integration of the mobility of passengers and goods (crowdshipping) [19,142,143].

Open innovation then plays a decisive role in the promotion of stakeholders' involvement in order to achieve sustainable growth for e-marketplaces and avoid the collapse of the ecosystems of these platforms [16,34,144] because the interaction with stakeholders influences the sustainability impact, and the degree of integration of different stakeholders will

help minimize negative social and ecological repercussions [105]. Therefore, e-marketplace platforms must understand the users' motives for engagement [104], and must encourage collaboration between stakeholders to access expertise, solve complex problems, and gain social legitimacy [16,88]. Likewise, these platforms must be prepared to receive feedback from customers [145], and to facilitate an open-innovation-friendly culture and open business model feedback loop, as Alibaba has done [35].

In order to achieve this, open innovation strategies (inward oriented and outward oriented) could determine new business models [28,38], and could assist in overcoming the barriers to sustainable online consumption [22], in such a way as to guarantee the sustainability of e-marketplaces, considering the points of view of the stakeholders. In this way, if the business model of an e-marketplace is properly structured whilst considering the users' needs, it will be possible to establish the gaps between the requirements of users (sellers and buyers) and the offer of services of the digital platform. Therefore, improvement opportunities could support a rapid response to the market, ensuring the sustainability of the e-marketplace [3]. Similarly, the business models of e-marketplaces must promote financial benefits and trust in other users in order to reduce the perceived risk and increase purchase intention [146], and ecological sustainability in order to promote the use of electronic platforms [104]. Consequently, Amazon has developed a disruptive business model by introducing new innovative fashion models such as Prime Wardrobe, AI Algo, Fashion designer, Echo Look, AR Mirror, Personal Shopper, Style Snap, and The Drop [144].

The review conducted in this study identifies the opportunity to develop instruments that measure open innovation with a sustainability approach, specifically in e-marketplaces, understood as digital platforms contributing to the reduction of communication and transaction costs by allowing many companies to sell their goods and services to other companies and consumers that are geographically distant. In this sense, the development of a conceptual framework that supports the measurement of open innovation with a sustainability approach will require the articulation of the pillars of sustainability [147], external stakeholders to the e-marketplace [30], inside-out and outside-in interactions [29], and the macro and micro dynamics of open innovation [54]. E-marketplaces constitute an opportunity for local small- and medium-sized enterprises (SMEs) to mitigate economic crises, as these companies are incapable of responding quickly to economic and financial recessions. In addition, e-marketplaces generate high user traffic, giving SMEs a greater chance of increasing their sales [148]. However, it is necessary to develop business models that consider the opinions and requirements of SMEs as suppliers/buyers in B2B models, as well as the requirements of consumers as buyers in B2C models. Therefore, research efforts should focus on the integration of stakeholders through open innovation strategies in order to obtain opportunities to improve business models and guarantee the social, environmental, and economic sustainability of e-marketplace platforms.

## 5. Conclusions

E-marketplaces represent digital platforms dedicated to the sale and marketing of goods and services in the era of business digitalization, which—by involving the interests of stakeholders through open innovation—allows the achievement of the sustainability of business models. Due to the importance of this topic, the publication of documents has increased, especially in the last five years. This study detected that the countries with the highest production on this subject are located mainly in Asia and Europe, with China standing out as the leading country in terms of publications in all of the periods analyzed, followed by countries such as India, Italy, Spain, the United Kingdom, and the United States. The main subject areas which most of the documents revolve around are social sciences, environmental sciences, energy, and business, management, and accounting, which involve the economic, environmental, and social dimensions of sustainability.

Within these subject areas, this study identified that the main concepts related to e-marketplaces, open innovation, and sustainability are sustainable development, e-commerce,

digital marketing, China, logistics, supply chain management, big data, planning, and decision making. Sustainable development represents the relationship between economic growth in e-marketplaces and the environment, considering stakeholders' collaboration for business model improvement. E-commerce stands out due to the rise of digital platforms that improve conditions for buyers and sellers, reducing environmental impacts related to mobility in cities and supporting the new dynamics caused by the COVID-19 pandemic. Digital marketing and sales focus on the promotion and commercialization of products in e-marketplaces. China represents the primary player in the growth of e-commerce and e-marketplaces. Logistics, supply chain management, and big data support the operation of electronic business models through the integration of partners and the management of transportation, distribution, and delivery operations. Planning and decision-making facilitate policy-making definition and policy deployment in order to achieve e-commerce sustainability.

Based on the results of this study, we suggest that authors researching e-marketplaces, open innovation, and sustainability publish their manuscripts in journals such as Sustainability Switzerland, the Journal of Open Innovation Technology Market and Complexity, and the Journal of Cleaner Production. These journals publish a significant number of articles per year on this topic, and they receive representative citations every year, generating a good impact in the scientific world. Future research should focus on solving traffic congestion, pollution, and emissions without causing higher costs for companies, whilst also reducing environmental impacts and achieving sustainability. Future works can address new delivery practices in last-mile logistics, and could comprehensively analyze crowdshipping solutions, considering legal frameworks in order to guarantee fair competition and job security for employees.

Likewise, this study identified research opportunities in the analysis of user needs, feedback from e-marketplace users, and the evolution of users' motives for engagement in e-marketplaces in order to adapt business models that fill gaps between the users' requirements (buyers, sellers, logistics operators) and the offer of electronic platforms. Therefore, we expect—in forthcoming studies—the development of reference frameworks to achieve sustainable growth in e-marketplaces by the integration of sustainability pillars and external stakeholders through open innovation, jointly addressing environmental, social, and economic aspects on both the retailer and consumer sides. Finally, this study encourages researchers to perform studies focused on SMEs, and the limitations and perceived benefits in the use of e-marketplaces, in order to adapt SMEs to the operational demands of e-commerce environments.

**Author Contributions:** Conceptualization, J.A.C. and A.L.-P.; methodology, J.A.C. and A.L.-P.; validation, M.F.C. and H.B.P.; formal analysis, J.A.C. and A.L.-P.; investigation, J.A.C. and A.L.-P.; resources, M.F.C. and H.B.P.; data curation, C.R. and T.A.; writing—original draft preparation, J.A.C.; writing—review and editing, J.A.C. and A.L.-P.; visualization, J.A.C., C.R. and T.A.; supervision, M.F.C. and H.B.P.; project administration, J.A.C. and A.L.-P.; funding acquisition, M.F.C. and H.B.P. All authors have read and agreed to the published version of the manuscript.

**Funding:** This research received no external funding.

**Data Availability Statement:** The data presented in this study are available on request from the corresponding author.

**Conflicts of Interest:** The authors declare no conflict of interest.

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
