# Peer review of "A Bibliometric Analysis and Systematic Review on E-Marketplaces, Open Innovation, and Sustainability"

_sustainability, doi:10.3390/su14095456_

Round 1

Reviewer 1 Report

The presented article is quite interesting paper dealing with good scientific topic. Its not very easy for the non-expert readers, since its full of economic and mathematical graphs and diagrams. On the other hand, its very interesting topic nowadays for the expert readers. 
Author presents methodological overview at the beginning of the article and all the scientific methods are appropriately used in the article. The scientific soundness of the topic mentioned in the article is good enough. The content of the article is good match with the topic of the journal, so the are no objections in this way. The citation and the list of references could be improved, because there is much more literature dealing with this topic, but its not major objection, which could cause the rejection of the article.

Author Response

Dear Bojana Radonjic                                                                                   Medellín, March 14th, 2022

Economics
MDPI

We communicate with you through this letter in order to outline every change made in the article sustainability-1626694 “A bibliometric analysis and systematic review on e-marketplaces, open innovation, and sustainability” based on the observations and suggestions provided by the Reviewer 1, Reviewer 2, and Reviewer 3. All changes and modifications made to the manuscript are highlighted in blue color in the revised version.

Comments from author to reviewers:
-Reviewer 1

The presented article is quite interesting paper dealing with good scientific topic. Its not very easy for the non-expert readers, since its full of economic and mathematical graphs and diagrams. On the other hand, its very interesting topic nowadays for the expert readers. Author presents methodological overview at the beginning of the article and all the scientific methods are appropriately used in the article. The scientific soundness of the topic mentioned in the article is good enough. The content of the article is good match with the topic of the journal, so the are no objections in this way.

R// We appreciate the reviewer's comments and consider the topic of the article to be a very interesting topic nowadays.

The citation and the list of references could be improved, because there is much more literature dealing with this topic, but its not major objection, which could cause the rejection of the article.

R// Following the reviewer's suggestions, we added 15 references to improve the list of references. These references belong to the year 2020, 2021, and 2022:

Hossain, I.; Azam, S.; Quaddus, M. Small firm entry to e ‑ marketplace for market expansion and internationalization: A theoretical perspective. J. Int. Entrep. 2021, 19, 560–590, doi:10.1007/s10843-021-00297-5.

Kusumawati, R.D.; Oswari, T.; Yusnitasari, T.; Dutt, H. Analysis of marketing mix and website performance on e-marketplace of agricultural products. Lect. Notes Networks Syst. 2022, 314, 437–444, doi:10.1007/978-981-16-5655-2_42.

Chandra, Y.U. Determinant factors of purchase intention in e-marketplace during covid-19 pandemic. ACM Int. Conf. Proceeding Ser. 2021, 45–48, doi:10.1145/3489088.3489099.

Dewi, K.C.; Ayuni, N.W.D. Business process re-enginering of tourism e-marketplace by engaging government , small medium enterprises and tourists. Bull. Electr. Eng. Informatics 2021, 10, 2866–2874, doi:10.11591/eei.v10i5.3159.

Inayatulloh, E.S. Private e-marketplace model for SMEs. In Proceedings of the Proceedings of 2021 International Conference on Information Management and Technology, ICIMTech 2021; 2021; pp. 481–484.

Christian, Y.; Utama, Y. Issues and determinant factors of customer feedback on e-commerce (e-marketplace). In Proceedings of the Proceedings of 2021 International Conference on Information Management and Technology, ICIMTech 2021; 2021; pp. 234–239.

Beraldi, P.; Maio, A. De; Laganà, D.; Violi, A. A pick-up and delivery problem for logistics e-marketplace services. Optim. Lett. 2021, 15, 1565–1577, doi:10.1007/s11590-019-01472-3.

Sedek, K.A.; Osman, M.N.; Omar, M.A.; Wahab, M.H.; Idrus, S.Z.S. Smart agro e-marketplace architectural model based on cloud data platform smart agro e-marketplace architectural model based on cloud data platform. J. Phys. Conf. Ser. 2021, 1874, 012022, doi:10.1088/1742-6596/1874/1/012022.

Oswari, T.; Kusumawati, R.D.; Yusnitasari, T.; Dutt, H.; Dunan, A. Factors influencing consumer intention in indonesia to utilize e-marketplace of agricultural products. In Proceedings of the Proceeding - 2021 2nd International Conference on ICT for Rural Development, IC-ICTRuDev 2021; 2021.

Alazab, K.; Dick, M.; Far, S.M. Assessing the effect of UTAUT2 on adoption of B2B/C2C e-marketplaces. J. Internet E-bus. Stud. 2020, 690228, doi:10.5171/2020.690228.

Malak, F.; Ferreira, J.B.; Pessoa, R.; Falcão, D.Q.; Malak, F.; Brantes, J.; Pessoa, R.; Falc, D.Q. Seller reputation within the B2C e-marketplace and impacts on purchase intention. Lat. Am. Bus. Rev. 2021, 22, 287–307, doi:10.1080/10978526.2021.1893182.

Croitor, E.; Werner, D.; Adam, M.; Benlian, A. Opposing effects of input control and clan control for sellers on e-marketplace platforms. Electron. Mark. 2021, doi:10.1007/s12525-021-00465-4.

Martins, N.; Brand, D.; Alvelos, H.; Silva, S. E-marketplace as a tool for the revitalization of portuguese craft industry: The design process in the development of an online platform. Futur. Internet 2020, 12, 1–23, doi:10.3390/fi1211019.

Chang, Y.; Lin, S.; Yen, D.C.; Hung, J. The trust model of enterprise purchasing for B2B e-marketplaces. Comput. Stand. Interfaces 2020, 70, 103422, doi:10.1016/j.csi.2020.103422.

Trott, M.; Baur, N.F.; Auf der Landwehr, M.; Rieck, J.; von Viebahn, C. Evaluating the role of commercial parking bays for urban stakeholders on last-mile deliveries – A consideration of various sustainability aspects. J. Clean. Prod. 2021, 312, 127462, doi:10.1016/j.jclepro.2021.127462

We added new references in the Introduction to support the relevance of a longitudinal study:

Battaglia, M.; Passetti, E.; Bianchi, L.; Frey, M. Managing for integration: a longitudinal analysis of management control for sustainability. J. Clean. Prod. 2016, 136, 213–225, doi:10.1016/j.jclepro.2016.01.108.

Takeuchi, M.; Meguro, A.; Yoshida, M.; Yamane, T.; Yatsu, K.; Okada, E.; Mizuki, N. Longitudinal analysis of 5 ‑ year refractive changes in a large Japanese population. Sci. Rep. 2022, 12, 2879, doi:10.1038/s41598-022-06898-x.

Similarly, new references were added to define the differences between sustainability and sustainable development, and between open innovation and innovation:

Hidalgo, A. Pensamiento económico sobre desarrollo; Universidad de Huelva: Huelva, Spain, 1998; ISBN 84-88751-62-1.

O’Riordan, T. The politics of sustainability. In Sustainable Environmental Management: Principles and Practice; Turner, K., Ed.; Westview Press: London, Belhaven, 1988; pp. 29–50.

Shaker, R.R. The spatial distribution of development in Europe and its underlying sustainability correlations. Appl. Geogr. 2015, 63, 304–314, doi:10.1016/j.apgeog.2015.07.009.

Londoño, A.; Cruz, J.G. Evaluation of sustainable development in the sub-regions of Antioquia (Colombia) using multi-criteria composite indices: A tool for prioritizing public investment at the subnational level. Environ. Dev. 2019, 1–22, doi:10.1016/j.envdev.2019.05.001.

Bernal-Torres, C.A.; Frost-González, S. Open innovation in Colombian enterprises: Challenge to overcome. Rev. Venez. Gerenc. 2015, 20, 252–267, doi:10.31876/revista.v20i70.19996.

Chesbrough, H. Open innovation: the new imperative for creating and profiting from technology; Harvard Business School Press: Boston, USA, 2003; ISBN 9781578518371.

Comments from author to reviewers:
-Reviewer 2

It was just for me, the article is too long, the author(s) may reduce it to 20-25 pages, if possible

R// We understand very well the position of the reviewer since the manuscript presents a length superior to the standard of scientific publications. In this case, the manuscript is of significant length because it introduces a longitudinal analysis of three periods, showing in each period the growth of documents, subject areas, leading journals, leading authors, scientific output by country/territory, leading affiliations, most cited documents, main concepts, and co-occurrence analysis.

Introduction is okay, but no justification on why this study focus on long, medium and short term 

R// We added a paragraph in the Introduction to justify why this study focuses on the long, medium, and short term (Line 189-197).

“Longitudinal studies are relevant for tracking changes over time in a specific field, assessing phenomena occurring over a long period, and describing how perspectives change over time [58,59]. Longitudinal studies applied to scientific literature analysis may require reviewing the changes presented in a research field in the last ten (long term), five (medium-term), and two (short term) years. Periods may not be consistent when performing longitudinal studies when considering the latest ten, five, and two years since notable growth of publications in the literature is usually generated in recent years (last two years). Likewise, an analysis of the previous two years allows identifying the latest trends, opportunities, and research gaps [57].”

Methodology needs some additional information about how classification of concepts/main topics were developed. And then, how these concepts is classified to the 6 clusters.

R// We added a paragraph in the Introduction to explain how clusters are created in VosViewer

(Lines 241-244).

“This software groups the concepts into clusters, and the number of clusters will depend on the number and type of concepts in each period. Each cluster has a central node gathering around many concepts, and they appear in large spheres in the co-occurrence graph.”

The title of this article mentioned three main concepts, e-market places, open innovation and sustainability. However, the results highlight the main concept is sustainable development, open innovation and electronic commerce, any justification about that?

R// The three main concepts addressed in this study (e-marketplaces, open innovation, and sustainability) were used in different combinations in the search equations in Scopus; therefore, it is expected that these concepts are relevant in the documents obtained. Other additional concepts are highlighted in Figure 3, Figure 6, Figure 9 for the long, medium, and short term. It could even be expected that these concepts would occupy the first three positions in these figures. However, the results show that the most relevant concept is sustainable development, which represents a reference framework that encompasses the central topic of the article.

Any clear definition about concept of sustainability versus sustainable development; the   electronic commerce versus e-commerce, the open innovation versus innovation? Please state.

R// We added a paragraph to explain the difference between sustainability and sustainable development (Line 666-673).

“Regarding the difference between sustainable development and sustainability, sustainable development focuses on the development concept [4], sustainability focuses on the environment concept [5]. Authors such as Shaker [6] suggest that sustainability represents the goal while sustainable development describes the process to achieve this goal. In a business context, corporate sustainability is measured by different international standards such as the Global Reporting Initiative (GRI) and the Dow Jones Sustainability (DJS), which evaluate different dimensions such as environmental, economic, and social [7].”

We added a paragraph to explain electronic commerce versus e-commerce (Line 674-676).

“The concepts of Electronic Commerce and E-commerce refer to the process of buying and selling products or services electronically [113], where E-commerce represents the represents the abbreviation of electronic commerce.”

We added a paragraph to explain the difference between open innovation and innovation (Line 683-692).

“Regarding the difference between innovation and open innovation, the traditional approach to innovation has been supported by the generation and development of ideas arising and treated from within the R&D departments [136], which is insufficient since it leads to depending on the creation of successful ideas and the hiring of experts in the business units that require innovation activities [29], which in turn results more expensive for a company. Open innovation requires both inside-out and outside-in interactions from the company, which is more efficient than one-way interaction [137]. Open innovation implies a dynamic company interaction with external stakeholders such as customers, users, suppliers, universities, research and development centers, competitors, government agencies, surrounding communities, and society [30].”

Any reason why sustainable development and sustainability is classified in different cluster/concept? The electronic commerce, e-commerce and commerce too. Please verify.

R// Sustainable development and sustainability, electronic commerce, e-commerce, and commerce are classified in different clusters due to the operating logic of the VosViewer software, which groups the keywords of the Scopus documents according to co-occurrence criteria. The denomination and context of these concepts depend on each document found in Scopus, which generates the clusters shown in Figure 3, Figure 6, Figure 9.

Any justification why cluster were developed into 6 clusters (lines 319-331); And, on how the concepts in Figure 3 is selected and linked to the appropriate cluster. The first sentence (line 319) is not clear. How the cluster is categorized into main node or central node? This is not clearly mentioned as well.

R// We added a paragraph in the Introduction to explain how clusters are created in VosViewer

(Lines 241-244).

“This software groups the concepts into clusters, and the number of clusters will depend on the number and type of concepts in each period. Each cluster has a central node gathering around many concepts, and they appear in large spheres in the co-occurrence graph.”

We added a phrase to explain how the concepts in Figure 3 are selected and linked to the appropriate cluster (Line 339-342)

“These clusters are identified using the item filter and zooming in the Vosviewer software to associate the colors of the nodes to each concept. The central node of each cluster is identified through the sphere size, font size, and the number of connections with other nodes.”

Please re-check the sentence, in Figure 10 (lines 554-555), the cluster 1 (red cluster) is referred to sustainability or sustainable development?

R// We verified the red cluster, which refers to the concept of Sustainability, and the sustainable development concept belongs to cluster 2 (green cluster). Due to VosViewer data visualization, the Sustainability concept is not printed in Figure 10; however, when detailing and enlarging the graph in the VosViewer software, it is confirmed that said node corresponds to Sustainability.

Figure 11- The title is not clear, whether the author (s) want to focus on concepts or on topics? Why not just classified it based on the clusters?

R// We corrected the title of Figure 3, Figure 6, Figure 9, and Figure 11:

Figure 3. Main concepts in 2012-2021.

Figure 6. Main concepts in 2017-2021.

Figure 9. Main concepts in 2020-2021.

Figure 11. Evolution of concepts by period.

Figure 11 shows the evolution over time of the main concepts that were presented in Figure 3, Figure 6, Figure 9, highlighting which concepts gain or lose relevance in time.

Some words in capital, and some with small letters (see line 578-579)

R// This observation was corrected by writing all subject areas in lower case throughout the manuscript.

References – need to be consistent in their formatting, either capital each word or sentence case.

R// We standardized the formatting of titles in the references, using capital letters only for the first words or proper names.

Comments from author to reviewers:
-Reviewer 3

Why the years chosen is not consistent? Revision is required.

R// We added a paragraph in the Introduction to justify why this study focuses on the long, medium, and short term (Line 189-197).

““Longitudinal studies are relevant for tracking changes over time in a specific field, assessing phenomena occurring over a long period, and describing how perspectives change over time [58,59]. Longitudinal studies applied to scientific literature analysis may require reviewing the changes presented in a research field in the last ten (long term), five (medium-term), and two (short term) years. Periods may not be consistent when performing longitudinal studies when considering the latest ten, five, and two years since notable growth of publications in the literature is usually generated in recent years (last two years). Likewise, an analysis of the previous two years allows identifying the latest trends, opportunities, and research gaps [57].”

Finally, we thank the reviewers for their suggestions and observations, which helped to improve the quality of the article, both in its form and content, ensuring that it is an article of high impact for the academic and scientific community.

Reviewer 2 Report

It was just for me, the article is too long, the author(s) may reduce it to 20-25 pages, if possible.

-Abstract is okay

-Introduction is okay, but no justification on why this study focus on long, medium and short term  

-Methodology needs some additional information about how classification of concepts/main topics were developed. And then, how these concepts is classified to the 6 clusters.

-Results are fine. The research objective is attainable. However, in my opinion, these aspects need some improvements;

  • The title of this article mentioned three main concepts, e-market places, open innovation and sustainability. However, the results highlight the main concept is sustainable development, open innovation and electronic commerce, any justification about that?
  • Any clear definition about concept of sustainability versus sustainable development; the   electronic commerce versus e-commerce, the open innovation versus innovation? Please state.
  • Any reason why sustainable development and sustainability is classified in different cluster/concept? The electronic commerce, e-commerce and commerce too. Please verify.
  • Any justification why cluster were developed into 6 clusters (lines 319-331); And, on how the concepts in Figure 3 is selected and linked to the appropriate cluster. The first sentence (line 319) is not clear.
  • How the cluster is categorized into main node or central node? This is not clearly mentioned as well.
  • Please re-check the sentence, in Figure 10 (lines 554-555), the cluster 1 (red cluster) is referred to sustainability or sustainable development?
  • Figure 11- The title is not clear, whether the author (s) want to focus on concepts or on topics? Why not just classified it based on the clusters?

Others:

  • Some words in capital, and some with small letters (see line 578-579)
  • References – need to be consistent in their formatting, either capital each word or sentence case.

Author Response

(The authors gave the same response as above.)

Reviewer 3 Report

 This study presents a bibliometric analysis and systematic review on e-marketplaces, open innovation, and sustainability for the last ten, five, and two years. Why the years chosen is not consistent? Revision is required.

Author Response

(The authors gave the same response as above.)
